# Deterministic Uncertainty Propagation for Improved Model-Based Offline Reinforcement Learning

**Abdullah Akgül**    **Manuel Haußmann**    **Melih Kandemir**
Department of Mathematics and Computer Science
University of Southern Denmark
Odense, Denmark
`{akgul,haussmann,kandemir}@imada.sdu.dk`

## Abstract

Current approaches to model-based offline reinforcement learning often incorporate uncertainty-based reward penalization to address the distributional shift problem. These approaches, commonly known as pessimistic value iteration, use Monte Carlo sampling to estimate the Bellman target to perform temporal difference-based policy evaluation. We find out that the randomness caused by this sampling step significantly delays convergence. We present a theoretical result demonstrating the strong dependency of suboptimality on the number of Monte Carlo samples taken per Bellman target calculation. Our main contribution is a deterministic approximation to the Bellman target that uses progressive moment matching, a method developed originally for deterministic variational inference. The resulting algorithm, which we call Moment Matching Offline Model-Based Policy Optimization (MOMBO), propagates the uncertainty of the next state through a nonlinear Q-network in a deterministic fashion by approximating the distributions of hidden layer activations by a normal distribution. We show that it is possible to provide tighter guarantees for the suboptimality of MOMBO than the existing Monte Carlo sampling approaches. We also observe MOMBO to converge faster than these approaches in a large set of benchmark tasks.

## 1   Introduction

Offline reinforcement learning  (Lange et al., 2012; Levine et al., 2020) aims to solve a control task using an offline dataset without interacting with the target environment. Such an approach is essential in cases where environment interactions are expensive or risky (Shortreed et al., 2011; Singh et al., 2020; Nie et al., 2021; Micheli et al., 2022). Directly adapting traditional online off-policy reinforcement learning methods to offline settings often results in poor performance due to the adverse effects of the distributional shift caused by the policy updates (Fujimoto et al., 2019; Kumar et al., 2019). The main reason for the incompatibility is the rapid accumulation of overestimated action-values during policy improvement. Pessimistic Value Iteration (PEVI) (Jin et al., 2021) offers a theoretically justified solution to this problem that penalizes the estimated rewards of synthetic state transitions proportionally to the uncertainty of the predicted next state. The framework encompasses many state-of-the-art offline reinforcement learning algorithms as special cases (Yu et al., 2020; Sun et al., 2023).

Model-based offline reinforcement learning approaches first fit a probabilistic model on the real state transitions and then supplement the real data with synthetic samples generated from this model throughout policy search (Janner et al., 2019). One line of work addresses distributional shift by constraining policy learning (Kumar et al., 2019; Fujimoto and Gu, 2021) where the policy is trained to mimic the behavior policy and is penalized based on the discrepancy between its actions and those

of the behavior policy, similarly to behavioral cloning. A second strategy introduces conservatism to training by *(i)* perturbing the training objective with a high-entropy behavior policy (Kumar et al., 2020; Yu et al., 2021), *(ii)* penalizing state-action pairs proportionally to their estimated variance (Yu et al., 2020; An et al., 2021; Bai et al., 2022; Sun et al., 2023), or *(iii)* adversarially training the environment model to minimize the value function (Rigter et al., 2022) to prevent overestimation in policy evaluation for out-of-domain state-action pairs.

In the offline reinforcement learning literature, uncertainty-driven approaches exist for both model-free (An et al., 2021; Bai et al., 2022) and model-based settings (Yu et al., 2020; Sun et al., 2023), all aimed at learning a pessimistic value function (Jin et al., 2021) by penalizing it with an uncertainty estimator. The impact of uncertainty quantification has been investigated in both online (Abbas et al., 2020) and offline (Lu et al., 2021) scenarios, particularly in model-based approaches, which is the focus of our work.

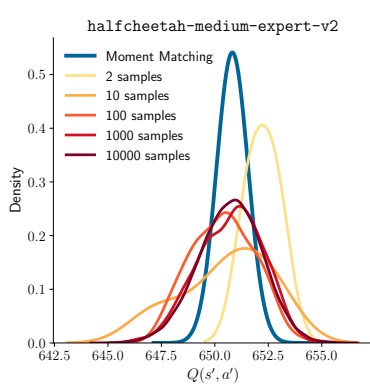

Despite the strong theoretical evidence suggesting that the quality of the Bellman target uncertainties has significant influence on model quality (O'Donoghue et al., 2018; Luis et al., 2023; Jin et al., 2021), the existing implementations rely on Monte Carlo samples and heuristics-driven uncertainty quantifiers being used as reward penalizers (Yu et al., 2020; Sun et al., 2023) which results in a high degree of randomness that manifests itself as significant delays in model convergence. Figure 1 demonstrates why this is the case by a toy example. The uncertainty around the estimated action-value of the next state does not shrink even after taking 10000 samples on the next state and passing them through a Q-network. Our moment matching-based approach predicts a similar mean with significantly smaller variance at the cost of only two Monte Carlo samples.

Figure 1: *Moment Matching versus Monte Carlo Sampling.* Moment matching offers sharp estimates of the action-value of the next state at the cost of only two forward passes through a critic network. A similar sharpness cannot be reached even with 10000 Monte Carlo samples, which is 5000 times more costly. See Appendix C.1.1 for details.

We identify the randomness introduced by Bellman target approximation via Monte Carlo sampling as the primary limiting factor for the performance of the PEVI approaches used in offline reinforcement learning. We have three main contributions:

*(i)* We present a new algorithm that employs progressive moment matching, an idea originally developed for deterministic variational inference of Bayesian neural networks (Wu et al., 2019a), for the first time to propagate deterministically environment model uncertainties through Q-function estimates. We refer to this algorithm as *Moment Matching Offline Model-Based Policy Optimization* (MOMBO).

*(ii)* We develop novel suboptimality guarantees for both MOMBO and existing sampling-based approaches highlighting that MOMBO is provably more efficient.

*(iii)* We present comprehensive experiment results showing that MOMBO significantly accelerates training convergence while maintaining asymptotic performance.

## 2 Preliminaries

**Reinforcement learning.** We define a Markov Decision Process (MDP) as a tuple $\mathcal{M} \triangleq (\mathcal{S}, \mathcal{A}, r, \mathrm{P}, \mathrm{P}_1, \gamma, H)$ where $\mathcal{S}$ represents the state space and $\mathcal{A}$ denotes the action space. We have $r : \mathcal{S} \times \mathcal{A} \rightarrow [0, R_{\max}]$ as a bounded deterministic reward function and $\mathrm{P} : \mathcal{S} \times \mathcal{A} \times \Delta(\mathcal{S}) \rightarrow [0, 1]$ as the probabilistic state transition kernel, where $\Delta(\mathcal{S})$ denotes the probability simplex defined over the state space. The MDP has an initial state distribution $\mathrm{P}_1 \in \Delta(\mathcal{S})$, a discount factor $\gamma \in (0, 1)$, and an episode length $H$. We use deterministic policies $\pi : \mathcal{S} \rightarrow \mathcal{A}$ and deterministic rewards in analytical demonstrations for simplicity and without loss of generality. Our results extend straightforwardly to probabilistic policies and reward functions. The randomness on the next state may result from the system stochasticity or the estimation error of a probabilistic model trained on the data collected from a deterministic environment. We define the action-value of a policy

$\pi$ for a state-action pair $(s, a)$ as

$$Q_\pi(s, a) \triangleq \mathbb{E}_\pi \left[ \sum_{i=h}^{H} \gamma^{i-h} r(s_i, a_i) \Big| s_h = s, a_h = a \right].$$

This function maps to the range $[0, {R_{\max}}/{1-\gamma}]$.

**Offline reinforcement learning** algorithms perform policy search using an offline collected dataset $\mathcal{D} = \{(s, a, r, s')\}$ from the target environment via a behavior policy $\pi_\beta$. The goal is to find a policy $\pi$ that minimizes the suboptimality, defined as $\mathtt{SubOpt}(\pi; s) \triangleq Q_{\pi^*}(s, \pi^*(s)) - Q_\pi(s, \pi(s))$ for an initial state $s$ and optimal policy $\pi^*$. For brevity, we omit the dependency on $s$ in $\mathtt{SubOpt}(\cdot)$. Model-based reinforcement learning approaches often train state transition probabilities and reward functions on the offline dataset by maximum likelihood estimation with an assumed density, modeling these as an ensemble of heteroscedastic neural networks (Lakshminarayanan et al., 2017). Mainstream model-based offline reinforcement learning methods adopt the Dyna-style (Sutton, 1990; Janner et al., 2019), which suggests training an off-the-shelf model-free reinforcement learning algorithm on synthetic data generated from the learned environment model $\widehat{\mathcal{D}}$ using initial states from $\mathcal{D}$. It has been observed that mixing minibatches collected from synthetic and real datasets improves performance in both online (Janner et al., 2019) and offline (Yu et al., 2020, 2021) settings.

**Scope and problem statement.** We focus on model-based offline reinforcement learning due to its superior performance over model-free methods (Yu et al., 2020; Kidambi et al., 2020; Yu et al., 2021; Rigter et al., 2022; Sun et al., 2023). The primary challenge in offline reinforcement learning is *distributional shift* caused by a limited coverage of the state-action space. When the policy search algorithm probes unobserved state-action pairs during training, significant value approximation errors arise. In standard supervised learning, such errors diminish as new observations are collected. However, in reinforcement learning, overestimation errors are exploited by the policy improvement step and accumulate, resulting in a phenomenon known as the *overestimation bias* (Thrun and Schwartz, 1993). Techniques developed to overcome this problem in the online setting such as min-clipping (Fujimoto et al., 2018) are insufficient for offline reinforcement learning. Algorithms that use reward penalization proportional to the estimated uncertainty of an unobserved next state are commonly referred to as *Pessimistic Value Iteration* (Yu et al., 2020; Sun et al., 2023). These algorithms are grounded in a theory that provides performance improvement guarantees by bounding the variance of next state predictions (Jin et al., 2021; Uehara and Sun, 2022). We demonstrate the theoretical and practical limitations of PEVI-based approaches and address them with a new solution.

**Bellman operators.** Both our analysis and the algorithmic solution build on a problem that arises from the inaccurate estimation of the Bellman targets in critic training. The error stems from the fact that during training the agent has access only to the *sample Bellman operator*,

$$\widehat{\mathbb{B}}_\pi Q(s, a, s') \triangleq r(s, a) + \gamma Q(s', \pi(s')),$$

where $s'$ is a Monte Carlo sample of the next state. However, the actual quantity of interest is the *exact Bellman operator* that is equivalent to the expectation of the sample Bellman operator,

$$\mathbb{B}_\pi Q(s, a) \triangleq \mathbb{E}_{s' \sim P(\cdot|s,a)} \left[ \widehat{\mathbb{B}}_\pi Q(s, a, s') \right].$$

## 3 Pessimistic value iteration algorithms

A pessimistic value iteration algorithm (Jin et al., 2021), denoted as $\mathbb{A}_{\mathtt{PEVI}}(\widehat{\mathbb{B}}_\pi^\Gamma Q, \widehat{P})$, performs Dyna-style model-based learning using the following *pessimistic sample Bellman target* for temporal difference updates during critic training:

$$\widehat{\mathbb{B}}_\pi^\Gamma Q(s, a, s') \triangleq \widehat{\mathbb{B}}_\pi Q(s, a, s') - \Gamma_{\widehat{P}}^Q(s, a),$$

where $\Gamma_{\widehat{P}}^Q(s, a)$ admits a learned state transition kernel $\widehat{P}$ and a value function $Q$ to map a state-action pair to a penalty score. Notably, this penalty term does not depend on an observed or previously sampled $s'$ as it quantifies the uncertainty around $s'$ using $\widehat{P}$. The rest follows as running an off-the-shelf model-free online reinforcement learning algorithm, for instance Soft Actor-Critic (SAC)

(Haarnoja et al., 2018), on a replay buffer that comprises a blend of real observations and synthetic data generated from $\widehat{P}$ using the recent policy of an intermediate training stage. We study the following two PEVI variants in this paper due to their representative value:

i) *Model-based Offline Policy Optimization (MOPO)* (Yu et al., 2020) directly penalizes the uncertainty of a state as inferred by the learned environment model $\Gamma_{\widehat{P}}^Q(s,a) \triangleq \text{var}_{\widehat{P}}[s']$. MOPO belongs to the family of methods where penalties are based on the uncertainty of the next state.

ii) *MOdel-Bellman Inconsistency penalized offLinE Policy Optimization (MOBILE)* (Sun et al., 2023) propagates the uncertainty of the next state to the Bellman target through Monte Carlo sampling and uses it as a penalty:

$$\Gamma_{\widehat{P}}^Q(s,a) \triangleq \widehat{\text{var}}_{s'\sim\widehat{P}}^N[\widehat{\mathbb{B}}_\pi Q(s,a,s')]$$

where $\widehat{\text{var}}_{s'\sim\widehat{P}}^N$ denotes the empirical variance of the quantity in brackets with respect to $N$ samples drawn i.i.d. from $\widehat{P}$. MOBILE represents the family of methods that penalize rewards based on the uncertainty of the Bellman target.

Both approaches approximate the mean of the Bellman target by evaluating the sample Bellman operator $\widehat{\mathbb{B}}_\pi Q(s,a,s')$ with a single $s'$ available either from real environment interaction within the dataset or a single sample taken from $\widehat{P}$. The following theorem establishes the critical role the penalty term plays in the model performance.

**Theorem 1** (*Suboptimality of PEVI (Jin et al., 2021)*). *For any $\pi$ derived with $\mathbb{A}_{PEVI}(\widehat{\mathbb{B}}_\pi^\Gamma Q, \widehat{P})$ that satisfies*

$$|\mathbb{B}_\pi Q(s,a) - \widehat{\mathbb{B}}_\pi Q(s,a,s')| \leq \Gamma_{\widehat{P}}^Q(s,a), \qquad \forall (s,a) \in \mathcal{S} \times \mathcal{A},$$

*with probability at least $1-\delta$ for some error tolerance $\delta \in (0,1)$ where $s' \sim P(\cdot|s,a)$, the following inequality holds for any initial state $s_1 \in \mathcal{S}$:*

$$\text{SubOpt}(\pi) \leq 2 \sum_{i=1}^{H} \mathbb{E}_{\pi^*}\left[\Gamma_{\widehat{P}}^Q(s_i,a_i)\Big|s_1\right].$$

When deep neural networks are used as value function approximators, calculating their variance becomes analytically intractable even for normally distributed inputs. Consequently, reward penalties $\Gamma_{\widehat{P}}^Q$ are typically derived from Monte Carlo samples, which are prone to high estimator variance (see Figure 1). Our key finding is that using high-variance estimates for reward penalization introduces three major practical issues in the training process of offline reinforcement learning algorithms:

*(i)* The information content of the distorted gradient signal shrinks, causing delayed convergence to the asymptotic performance.

*(ii)* The first two moments of the Bellman target are poorly approximated for feasible sample counts.

*(iii)* Larger confidence radii need to be used to counter the instability caused by *(i)* and the high overestimation risk caused by the inaccuracy described in *(ii)*, which unnecessarily restricts model capacity.

### 3.1 Theoretical analysis of sampling-based PEVI

We motivate the benefit of our deterministic uncertainty propagation scheme by demonstrating the prohibitive relationship of the sample Bellman operator when used in the PEVI context. The analysis of the distribution around the Bellman operator can be characterized as follows. We are interested in the distribution of a deterministic map $y = f(x)$ for a normally distributed input $x \sim \mathcal{N}(\mu, \sigma^2)$. We analyze this complex object via a surrogate of it. For a sample set $S_N = \{x_i\}_{i=1}^N$ obtained i.i.d. from $\mathcal{N}(\mu, \sigma^2)$, let $\mu_N$ and $\sigma_N^2$ be its empirical mean and variance. Now consider the following two random variables $y_N = \frac{1}{N} \sum_{i=1}^N f(x_i)$ and $y_N' = f(x')$ for $x' \sim \mathcal{N}(\mu_N, \sigma_N^2)$. Note that $y_1$

and $y_1'$ are equal in distribution. Furthermore, both $y_N$ and $y_N'$ converge to the true $y$ as $N$ tends to infinity. We perform our analysis on $y_N'$ where analytical guarantees are easier to build. Furthermore, $y_N'$ is a tighter approximation of both $y_1$ and $y_1'$. We construct the following suboptimality proof for the PEVI algorithmic family that approximates the uncertainty around the Bellman operator in the way $y_N'$ is built. In this theorem and elsewhere, $A_l$ stands for the weights of the $l$-th layer of a Multi-Layer Perceptron (MLP) which has 1-Lipschitz activation functions and $\|\cdot\|$ denotes $L1$-norm. See Appendix B.2 for the proof.

**Theorem 2** (*Suboptimality of sampling-based PEVI*). *For any policy $\pi_{MC}$ learned by $\mathbb{A}_{PEVI}(\widehat{\mathbb{B}}_\pi^\Gamma Q, \widehat{P})$ using $N$ Monte Carlo samples to approximate the Bellman target with respect to an action-value network defined as an $L$-layer MLP with $1$-Lipschitz activation functions, the following inequality holds for any error tolerance $\delta \in (0, 1)$ with probability at least $1 - \delta$*

$$SubOpt(\pi_{MC}) \le 2H \prod_{l=1}^{L} \|A_l\| \sqrt{-\frac{8 \log(\delta/4) R_{\max}^2}{\lfloor N/2 \rfloor (1 - \gamma)^2}}.$$

The bound in Theorem 2 is prohibitively loose since $R_{\max}^2/(1 - \gamma)^2$ is a large number in practice. For instance, the maximum realizable step reward in the `HalfCheetah-v4` environment of the MuJoCo physics engine (Todorov et al., 2012) is at least around 11.7 according to Hui et al. (2023). Choosing the usual $\gamma = 0.99$, we obtain $R_{\max}^2/(1 - \gamma)^2 = 1.37 \times 10^6$. Furthermore, as the bound depends on the number of samples for the next state $N$, it becomes looser as $N \to 1$. The bound is not defined for $N = 1$, but the loosening trend is clear.

# 4 MOMBO: Moment matching offline model-based policy optimization

Our key contribution is the finding that quantifying the uncertainty around the Bellman target brings significant benefits to the model-based offline reinforcement learning setting. We obtain a deterministic approximation using a moment matching technique originally developed for Bayesian inference. This method propagates the estimated uncertainty of the next state $s'$ obtained from a learned environment model $\widehat{P}$ through an action-value network by alternating between an affine transformation of a normally distributed input to another normally distributed linear activation and projecting the output of a nonlinear activation to a normal distribution by analytically calculating its first two moments. The resulting normal distributed output is then used to build a lower confidence bound on the Bellman target, which is an equivalent interpretation of reward penalization. Such a deterministic approximation is both a more accurate uncertainty quantifier and a more robust quantity to be used during training in a deep reinforcement learning setting. Its contribution to robust and sample-efficient training has been observed in other domains (Wang and Manning, 2013; Wu et al., 2019a). Prior work attempted to propagate full covariances through deep neural nets (see, e.g. Wu et al., 2019a; Look et al., 2023; Wright et al., 2024) at a prohibitive computational cost (quadratic in the number of neurons) that does not bring a commensurate empirical benefit. Therefore, we choose to propagate only means and variances.

Assuming the input of a fully-connected layer to be $X \sim \mathcal{N}(X|\mu, \sigma^2)$, the pre-activation $Y$ for a neuron associated with weights $\theta$ is given by $Y = \theta^\top X$, i.e., $Y \sim \mathcal{N}(Y|\theta^\top \mu, (\theta^2)^\top \sigma^2)$, where we absorb the bias into $\theta$ for simplicity and the square on $\theta$ is applied element-wise. For a ReLU activation function $r(x) \triangleq \max(0, x)$,[1] mean $\widetilde{\mu}$ and variance $\widetilde{\sigma}^2$ of $Y = \max(0, X)$ are analytically tractable (Frey and Hinton, 1999). We approximate the output with a normal distribution $\widetilde{X} \sim \mathcal{N}(\widetilde{X}|\widetilde{\mu}, \widetilde{\sigma}^2)$ and summarize several properties regarding $\widetilde{\mu}$ and $\widetilde{\sigma}^2$ below. See Appendix B.1 for proofs of all results in this subsection.

**Lemma 1** (*Moment matching*). *For $X \sim \mathcal{N}(X|\mu, \sigma^2)$ and $Y = \max(0, X)$, we have*

$$(i) \quad \widetilde{\mu} \triangleq \mathbb{E}[Y] = \mu \Phi(\alpha) + \sigma \phi(\alpha), \qquad \widetilde{\sigma}^2 \triangleq \text{var}[Y] = (\mu^2 + \sigma^2)\Phi(\alpha) + \mu\sigma\phi(\alpha) - \widetilde{\mu}^2,$$

*where $\alpha = \mu/\sigma$, and $\phi(\cdot)$, $\Phi(\cdot)$ are the probability density function (pdf) and cumulative distribution function (cdf) of the standard normal distribution, respectively. Additionally, we have that*

$$(ii) \quad \widetilde{\mu} \ge \mu \quad and \quad (iii) \quad \widetilde{\sigma}^2 \le \sigma^2.$$

---

[1] Although we build our algorithm with the ReLU activation function, it is also applicable to other activation functions whose first and second moments are tractable or can be approximated with sufficient accuracy, including all commonly used activation functions.

Algorithm 1 outlines the moment matching process. This process involves two passes through the linear layer: one for the first moment and one for the second, along with additional operations that take negligible computation time. In contrast, sampling-based methods require $N$ forward passes, where $N$ is the number of samples. Thus, for any $N > 2$, sampling-based methods introduce computational overhead. MOBILE (Sun et al., 2023), e.g., uses $N = 10$. See Figure 1 and Figure 3 for illustrations on the effect of varying $N$.

---

**Algorithm 1** Deterministic uncertainty propagation through moment matching

> **function** MOMENTMATCHINGTHROUGHLINEAR($\theta, b, X$)
>     $\theta$: Weights of the layer, $b$: bias of the layer
>     $X = \mathcal{N}(X|\mu_X, \sigma_X^2)$                            ▷ input a normal distribution
>     $(\mu_Y, \sigma_Y^2) \leftarrow (\theta^\top \mu_X + b, \theta^{2\top} \sigma_X^2)$              ▷ transform mean and variance
>     **return** $Y = \mathcal{N}(Y|\mu_Y, \sigma_Y^2)$           ▷ output the transformed distribution
> **end function**
> **function** MOMENTMATCHINGTHROUGHRELU($X$)
>     $X = \mathcal{N}(X|\mu_X, \sigma_X^2)$                            ▷ input a normal distribution
>     $\alpha \leftarrow \mu_X / \sigma_X$
>     $\mu_Y \leftarrow \mu_X \Phi(\alpha) + \sigma_X \phi(\alpha)$          ▷ compute the first two moments of ReLU($X$)
>     $\sigma_Y^2 \leftarrow \left(\mu_X^2 + \sigma_X^2\right) \Phi(\alpha) + \mu_X \sigma_X \phi(\alpha) - \mu_Y^2$      ▷ $\phi/\Phi$ are the normal pdf/cdf
>     **return** $Y = \mathcal{N}(Y|\mu_Y, \sigma_Y^2)$       ▷ output a normal distribution with these moments
> **end function**

---

We propagate the distribution of the next state predicted by the environment model through the action-value function network using moment matching. We define a *moment matching Bellman target distribution* as:

$$\widetilde{\mathbb{B}}_\pi Q(s, a, s') \overset{d}{=} r(s, a) + \gamma Q_{MM}(s', \pi(s'))$$

for $s' \sim \widehat{\mathrm{P}}(\cdot | s, a)$ and $Q_{MM}(s', a')$ a normal distribution with mean $\mu_{MM}(s', a')$ and variance $\sigma_{MM}^2(s', a')$ obtained as the outcome of a progressive application of Algorithm 1 through the layers of a critic network $Q$. The sign $\overset{d}{=}$ denotes equality in distribution, i.e., the expressions on the two sides share the same probability law. We perform pessimistic value iteration using a lower confidence bound on $\widetilde{\mathbb{B}}_\pi Q(s, a, s')$ as the Bellman target

$$\widetilde{\mathbb{B}}_\pi^\Gamma Q(s, a, s') \triangleq r(s, a) + \gamma \mu_{MM}(s', \pi(s')) - \beta \gamma \sigma_{MM}(s', \pi(s')) \tag{1}$$

for some radius $\beta > 0$.

### 4.1 Theoretical analysis of moment matching-based uncertainty propagation

The following lemma provides a bound on the 1-Wasserstein distance $W_1$ between the true distribution $\rho_Y$ of a random variable after being transformed by a ReLU activation function and its moment matched approximation $\rho_{\widetilde{X}}$. See Appendix B.3 for the proofs of all results presented in this subsection.

**Lemma 2** (*Moment matching bound*)**.** *For the following three random variables*

$$X \sim \rho_X, \qquad Y = \max(0, X), \qquad \widetilde{X} \sim \mathcal{N}(\widetilde{X} | \widetilde{\mu}, \widetilde{\sigma}^2)$$

*with $\widetilde{\mu} = \mathbb{E}[Y]$ and $\widetilde{\sigma}^2 = \mathrm{var}[Y]$ the following inequality holds*

$$W_1(\rho_Y, \rho_{\widetilde{X}}) \le \int_{-\infty}^0 F_{\widetilde{X}}(u) du + W_1(\rho_X, \rho_{\widetilde{X}})$$

*where $F_{\widetilde{X}}(\cdot)$ is cdf of $\widetilde{X}$. If $\rho_X = \mathcal{N}(X|\mu, \sigma^2)$, it can be further simplified to*

$$W_1(\rho_Y, \rho_{\widetilde{X}}) \le \widetilde{\sigma} \phi\left(\frac{\widetilde{\mu}}{\widetilde{\sigma}}\right) - \widetilde{\mu} \Phi\left(-\frac{\widetilde{\mu}}{\widetilde{\sigma}}\right) + |\mu - \widetilde{\mu}| + |\sigma - \widetilde{\sigma}|.$$

Applying this to a moment matching $L$-layer MLP yields the following deterministic bound.

**Lemma 3** (*Moment matching MLP bound*). *Let $f(X)$ be an $L$-layer MLP with ReLU activation $r(x) = \max(0, x)$. For $l = 1, \ldots, L - 1$, the sampling-based forward-pass is*

$$Y_0 = X_s, \qquad Y_l = r(f_l(Y_{l-1})), \qquad Y_L = f_L(Y_{L-1})$$

*where $f_l(\cdot)$ is the $l$-th layer and $X_s$ a sample of $\mathcal{N}(X|\mu_X, \sigma_X^2)$. Its moment matching pendant is*

$$\widetilde{X}_0 \sim \mathcal{N}(\widetilde{X}_0|\mu_X, \sigma_X^2), \qquad \widetilde{X}_l \sim \mathcal{N}\left(\widetilde{X}_l \Big| \mathbb{E}\left[r(f_l(\widetilde{X}_{l-1}))\right], \text{var}\left[r(f_l(\widetilde{X}_{l-1}))\right]\right).$$

*The following inequality holds for $\widetilde{\rho}_Y = \rho_{\widetilde{X}_L} = \mathcal{N}(\widetilde{X}_L|\mathbb{E}[f(\widetilde{X}_{L-1})], \text{var}[f(\widetilde{X}_{L-1})])$.*

$$W_1(\rho_Y, \widetilde{\rho}_Y) \leq \sum_{l=2}^{L} \left(G(\widetilde{X}_{l-1}) + C_{l-1}\right) \prod_{j=l}^{L} \|A_j\|,$$

*with* $\quad G(\widetilde{X}_l) = \widetilde{\sigma}_l \phi\left(\frac{\widetilde{\mu}_l}{\widetilde{\sigma}_l}\right) - \widetilde{\mu}_l \Phi\left(-\frac{\widetilde{\mu}_l}{\widetilde{\sigma}_l}\right) \leq 1, \qquad C_l \leq |A_l \widetilde{\mu}_{l-1} - \widetilde{\mu}_l| + \left|\sqrt{A_l^2 \widetilde{\sigma}_{l-1}^2} - \widetilde{\sigma}_l\right|$

*where $\sqrt{\cdot}$ and $(\cdot)^2$ are applied elementwise.*

Relying on Lemma 6 again, this result provides an upper bound on the suboptimality of our moment matching approach.

**Theorem 3** (*Suboptimality of moment matching-based PEVI algorithms*). *For any policy $\pi_{MM}$ derived with $\mathbb{A}_{PEVI}(\widetilde{\mathbb{B}}_\pi^\Gamma Q, \widehat{P})$ learned by a penalization algorithm that uses moment matching to approximate the Bellman target with respect to an action-value network defined as an $L$-layer MLP with 1-Lipschitz activation functions, the following inequality holds*

$$SubOpt(\pi_{MM}) \leq 2H \sum_{l=2}^{L} \left(G(\widetilde{X}_{l-1}) + C_{l-1}\right) \prod_{j=l}^{L} \|A_j\|.$$

The bound in Theorem 3 is much tighter than Theorem 2 in practice as $R_{\max}^2/(1-\gamma)^2$ is very large while the Lipschitz continuities $\|A_j\|$ of the two-layer MLPs used in the experiments are in the order of low hundreds according to empirical investigations (Khromov and Singh, 2024). Another favorable property of Theorem 3 is that its statement is exact, as opposed to the probabilistic statement made in Theorem 2. The provable efficiency of MOMBO could be of independent interest for safety-critical use cases of offline reinforcement learning where the availability of analytical performance guarantees is a fundamental requirement.

## 4.2 Implementation details of MOMBO

We adopt the model learning scheme from Model-Based Policy Optimization (MBPO) (Janner et al., 2019) and use SAC (Haarnoja et al., 2018) as the policy search algorithm. These approaches represent the best practices in many recent model-based offline reinforcement learning methods (Yu et al., 2020, 2021; Sun et al., 2023). However, we note that most of our findings are more broadly applicable. Following MBPO, we train an independent heteroscedastic neural ensemble of transition kernels modeled as Gaussian distributions over the next state and reward. We denote each ensemble element as $\widehat{P}_n(s', r|s, a)$ for $n \in \{1, \ldots, N_{\text{ens}}\}$. After evaluating the performance of each model on a validation set, we select the $N_{\text{elite}}$ best-performing ensemble elements for further processing. We use these elite models to generate $k$-step rollouts with the current policy and to create the synthetic dataset $\widehat{\mathcal{D}}$, which we then combine with the real observations $\mathcal{D}$. We retain the mean and variance of the predictive distribution for the next state to propagate it through the action-value function while assigning zero variance to the real tuples.

The lower confidence bound given in Equation (1) builds on our MDP definition, which assumes a deterministic reward function and a deterministic policy for illustrative purposes. In our implementation, the reward model also follows a heteroscedastic ensemble of normal distributions. We also incorporate the uncertainty around the predicted reward of a synthetic sample into the Bellman target calculation. Furthermore, our policy function is a squashed Gaussian distribution trained by SAC in the maximum entropy reinforcement learning setting. For further details, refer to the Appendix C.2.

Table 1: *Performance evaluation on the D4RL dataset.* Normalized reward at 3M gradient steps and Area Under the Learning Curve (AULC) (mean±std) scores are averaged across four repetitions. The highest means are highlighted in bold and are underlined if they fall within one standard deviation of the best score. The average normalized score is the average across all tasks. The average ranking is based on the rank of the mean.

| Dataset Type | Environment | NORMALIZED REWARD (↑) | | | AULC (↑) | | |
|---|---|---|---|---|---|---|---|
| | | MOPO | MOBILE | MOMBO | MOPO | MOBILE | MOMBO |
| random | halfcheetah | $37.2_{\pm1.6}$ | $41.2_{\pm1.1}$ | $\mathbf{43.6_{\pm1.1}}$ | $36.3_{\pm1.0}$ | $39.5_{\pm1.2}$ | $\mathbf{41.4_{\pm1.0}}$ |
| | hopper | $\mathbf{31.7_{\pm0.1}}$ | $31.3_{\pm0.1}$ | $25.4_{\pm10.2}{}^{\dagger}$ | $\mathbf{28.6_{\pm1.4}}$ | $23.6_{\pm3.7}$ | $17.3_{\pm1.3}$ |
| | walker2d | $8.2_{\pm5.6}$ | $\mathbf{22.1_{\pm0.9}}$ | $21.5_{\pm0.1}$ | $5.4_{\pm3.2}$ | $18.0_{\pm0.4}$ | $\mathbf{19.2_{\pm0.5}}$ |
| | Average on random | 25.7 | **31.5** | 30.2 | 23.4 | **27.1** | 26.0 |
| medium | halfcheetah | $72.4_{\pm4.2}$ | $75.8_{\pm0.8}$ | $\mathbf{76.1_{\pm0.8}}$ | $70.9_{\pm2.0}$ | $72.1_{\pm1.0}$ | $\mathbf{73.0_{\pm0.9}}$ |
| | hopper | $62.8_{\pm38.1}$ | $103.6_{\pm1.0}$ | $\mathbf{104.2_{\pm0.5}}$ | $37.0_{\pm15.3}$ | $82.2_{\pm7.3}$ | $\mathbf{95.9_{\pm2.5}}$ |
| | walker2d | $85.4_{\pm2.9}$ | $\mathbf{88.3_{\pm2.5}}$ | $86.4_{\pm1.2}$ | $77.6_{\pm1.3}$ | $79.0_{\pm1.3}$ | $\mathbf{84.0_{\pm1.1}}$ |
| | Average on medium | 73.6 | **89.3** | 88.9 | 61.8 | 77.8 | **84.3** |
| medium-replay | halfcheetah | $\mathbf{72.1_{\pm3.8}}$ | $71.9_{\pm3.2}$ | $72.0_{\pm4.3}$ | $68.4_{\pm4.7}$ | $67.9_{\pm2.8}$ | $68.7_{\pm3.9}$ |
| | hopper | $92.7_{\pm20.7}$ | $\mathbf{105.1_{\pm1.3}}$ | $104.8_{\pm1.0}$ | $81.7_{\pm4.6}$ | $78.7_{\pm4.0}$ | $\mathbf{87.3_{\pm2.0}}$ |
| | walker2d | $85.9_{\pm5.3}$ | $\mathbf{90.5_{\pm1.7}}$ | $89.6_{\pm3.8}$ | $65.3_{\pm12.7}$ | $79.9_{\pm4.3}$ | $\mathbf{80.8_{\pm5.6}}$ |
| | Average on medium-replay | 83.4 | **89.2** | 88.8 | 72.4 | 75.5 | **78.9** |
| medium-expert | halfcheetah | $83.6_{\pm12.5}$ | $100.9_{\pm1.5}$ | $\mathbf{103.3_{\pm0.8}}$ | $77.1_{\pm4.0}$ | $94.5_{\pm1.8}$ | $\mathbf{95.2_{\pm0.7}}$ |
| | hopper | $74.9_{\pm44.2}$ | $112.5_{\pm0.2}$ | $\mathbf{112.6_{\pm0.3}}$ | $55.6_{\pm17.3}$ | $82.7_{\pm7.3}$ | $\mathbf{84.3_{\pm4.7}}$ |
| | walker2d | $108.2_{\pm4.3}$ | $\mathbf{114.5_{\pm2.2}}$ | $113.9_{\pm0.9}$ | $88.3_{\pm6.3}$ | $94.3_{\pm0.9}$ | $\mathbf{98.9_{\pm3.3}}$ |
| | Average on medium-expert | 88.9 | 109.3 | **109.9** | 73.6 | 90.5 | **92.8** |
| | Average Score | 67.6 | **79.8** | 79.5 | 57.5 | 67.7 | **70.5** |
| | Average Ranking | 2.7 | **1.7** | **1.7** | 2.7 | 2.2 | **1.2** |

$^{\dagger}$ *High standard deviation due to failure in one repetition, which can be mitigated by increasing $\beta$. Median result:* $31.3$

# 5 Experiments

We compare MOMBO against MOPO and MOBILE, the two representative PEVI variants, across twelve tasks from the D4RL dataset (Fu et al., 2020), which consists of data collected from three MuJoCo environments (halfcheetah, hopper, walker2d) with behavior policies exhibiting four degrees of expertise (random, medium, medium-replay, and medium-expert). We use the datasets collected with 'v2' versions of the MuJoCo environments to be commensurate with the state of the art. We evaluate model performance using two scores:

(i) *Normalized Reward:* Total episode reward collected in evaluation mode after offline training ends, normalized by the performances of random and expert policies.

(ii) *Area Under Learning Curve (AULC):* Average normalized reward computed at the intermediate steps of training.

AULC indicates how fast a model converges to its final performance. A higher AULC reflects more efficient learning other things being equal. We report the main results in Table 1 and provide the learning curves of the halfcheetah environment in Figure 2 for visual investigation. We present the learning curves for the remaining tasks in Figure 4. We obtain the results for the baseline models from the log files provided by the OfflineRL library (Sun, 2023). We implement MOMBO into the MOBILE (Sun et al., 2023) code base shared by its authors and use their experiment configurations wherever applicable. See Appendix C for details. The source code of our algorithm is available at https://github.com/adinlab/MOMBO. The OfflineRL library does not contain the log files for the random datasets for MOPO at the time of writing. We replicate these experiments using the MOBILE code based on the prescribed configurations.

Theorem 1 indicates that the performance of a PEVI algorithm depends on how tightly its reward penalizer $\Gamma_{\widehat{P}}^{Q}(s,a)$ upper bounds the Bellman operator error $|\mathbb{B}_{\pi}Q(s,a) - \widehat{\mathbb{B}}_{\pi}Q(s,a)|$. We use this theoretical result to compare the quality of the reward penalizers of the three models based on average values of the following two performance scores calculated across data points observed during ten evaluation episodes:

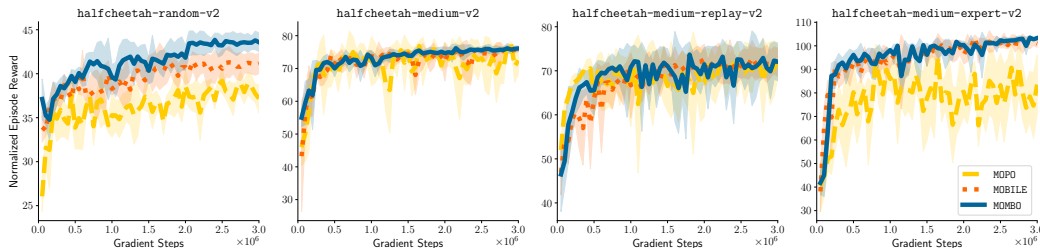

Figure 2: Evaluation results on `halfcheetah` for four settings. The dashed, dotted, and solid curves represent the mean of the normalized rewards across ten evaluation episodes and four random seeds. The shaded area indicates one standard deviation from the mean.

Table 2: *Uncertainty quantification on the D4RL dataset.* Accuracy and tightness (mean±std) scores are averaged across four repetitions. The highest means are highlighted in bold and are underlined if they fall within one standard deviation of the best score.

| Dataset Type | Environment | ACCURACY (↑) | | | TIGHTNESS (↑) | | |
|---|---|---|---|---|---|---|---|
| | | MOPO | MOBILE | MOMBO | MOPO | MOBILE | MOMBO |
| random | halfcheetah | $0.02_{\pm 0.0}$ | $\mathbf{0.25}_{\pm \mathbf{0.02}}$ | $\underline{0.24}_{\pm 0.07}$ | $-14.58_{\pm 0.63}$ | $-6.88_{\pm 0.61}$ | $\mathbf{-6.21}_{\pm \mathbf{0.28}}$ |
| | hopper | $0.14_{\pm 0.04}$ | $\mathbf{0.85}_{\pm \mathbf{0.03}}$ | $0.61_{\pm 0.07}$ | $-1.68_{\pm 0.22}$ | $\underline{-0.64}_{\pm 0.2}$ | $\mathbf{-0.31}_{\pm \mathbf{0.65}}$ |
| | walker2d | $0.01_{\pm 0.01}$ | $0.55_{\pm 0.04}$ | $\mathbf{0.74}_{\pm \mathbf{0.06}}$ | $-17 \cdot 10^{7}_{\pm 16 \cdot 10^{7}}$ | $-0.27_{\pm 0.08}$ | $\mathbf{-0.14}_{\pm \mathbf{0.02}}$ |
| medium | halfcheetah | $0.06_{\pm 0.01}$ | $0.25_{\pm 0.0}$ | $\mathbf{0.34}_{\pm \mathbf{0.04}}$ | $-15.71_{\pm 0.68}$ | $-10.03_{\pm 0.53}$ | $\mathbf{-9.06}_{\pm \mathbf{0.26}}$ |
| | hopper | $0.04_{\pm 0.01}$ | $0.41_{\pm 0.01}$ | $\mathbf{0.55}_{\pm \mathbf{0.03}}$ | $-4.16_{\pm 1.24}$ | $-3.14_{\pm 0.08}$ | $\mathbf{-1.3}_{\pm \mathbf{0.21}}$ |
| | walker2d | $0.02_{\pm 0.0}$ | $0.16_{\pm 0.01}$ | $\mathbf{0.38}_{\pm \mathbf{0.02}}$ | $-8.91_{\pm 0.61}$ | $-5.02_{\pm 0.52}$ | $\mathbf{-4.03}_{\pm \mathbf{0.24}}$ |
| medium-replay | halfcheetah | $0.09_{\pm 0.01}$ | $0.04_{\pm 0.0}$ | $\mathbf{0.16}_{\pm \mathbf{0.0}}$ | $-11.67_{\pm 0.94}$ | $-11.85_{\pm 0.41}$ | $\mathbf{-10.4}_{\pm \mathbf{0.66}}$ |
| | hopper | $0.02_{\pm 0.01}$ | $0.03_{\pm 0.01}$ | $\mathbf{0.17}_{\pm \mathbf{0.01}}$ | $-5.35_{\pm 0.71}$ | $-3.4_{\pm 0.04}$ | $\mathbf{-3.17}_{\pm \mathbf{0.04}}$ |
| | walker2d | $0.08_{\pm 0.01}$ | $0.14_{\pm 0.01}$ | $\mathbf{0.36}_{\pm \mathbf{0.02}}$ | $-4.47_{\pm 0.43}$ | $-4.56_{\pm 0.13}$ | $\mathbf{-3.78}_{\pm \mathbf{0.39}}$ |
| medium-expert | halfcheetah | $0.13_{\pm 0.02}$ | $0.36_{\pm 0.03}$ | $\mathbf{0.44}_{\pm \mathbf{0.03}}$ | $-21.25_{\pm 2.87}$ | $-13.22_{\pm 0.47}$ | $\mathbf{-11.88}_{\pm \mathbf{0.5}}$ |
| | hopper | $0.04_{\pm 0.02}$ | $0.43_{\pm 0.02}$ | $\mathbf{0.51}_{\pm \mathbf{0.04}}$ | $-9.77_{\pm 7.24}$ | $-3.5_{\pm 0.03}$ | $\mathbf{-3.38}_{\pm \mathbf{0.02}}$ |
| | walker2d | $0.05_{\pm 0.01}$ | $\mathbf{0.47}_{\pm \mathbf{0.02}}$ | $\underline{0.45}_{\pm 0.02}$ | $-9.77_{\pm 0.02}$ | $-5.52_{\pm 0.36}$ | $\mathbf{-5.29}_{\pm \mathbf{0.14}}$ |

(i) *Accuracy:* $\mathbb{1}(\Gamma^{Q}_{\widehat{\mathbb{P}}}(s,a) \geq |\mathbb{B}_{\pi}Q(s,a) - \widehat{\mathbb{B}}_{\pi}Q(s,a)|)$ for the indicator function $\mathbb{1}$, i.e., how often the reward penalizer is a valid $\xi-$uncertainty quantifier as assumed by Theorem 1.

(ii) *Tightness:* $\Gamma^{Q}_{\widehat{\mathbb{P}}}(s,a) - |\mathbb{B}_{\pi}Q(s,a) - \widehat{\mathbb{B}}_{\pi}Q(s,a)|$, i.e., how sharp a $\xi-$uncertainty quantifier the reward penalizer is.

Table 2 shows that MOMBO provides more precise uncertainty estimates compared to the sampling-based approaches. It also indicates that MOMBO provides tighter estimates of the Bellman operator error, meaning that the sampling-based approaches use larger confidence radii. See Appendix C.1.2 for further details.

The D4RL dataset is a heavily studied benchmark database where many hyperparameters, such as penalty coefficients, are tuned by trial and error. We argue that this is not feasible in a realistic offline reinforcement learning setting where the central assumption is that policy search needs to be performed without real environment interactions. Furthermore, it is more realistic to assume that collecting data from expert policies is more expensive than random exploration. One would then need to perform offline reinforcement learning on a dataset that comprises observations obtained at different expertise levels. The mixed offline reinforcement learning dataset (Hong et al., 2023) satisfies both desiderata, as the baseline models are not engineered for its setting and its tasks comprise data from two policies: an expert or medium policy and a random policy, presented in varying proportions. We compare MOMBO against MOBILE and MOPO on this dataset for a fixed and shared penalty coefficient for all models. We find that MOMBO outperforms both baselines in final performance and learning speed in this more realistic setup. See Appendix C.1.3 and Table 3 for further details and results.

Our key experimental findings can be summarized as follows:

(i) *MOMBO converges faster and trains more stably.* It learns more rapidly and reaches its final performance earlier as visible from the AULC scores. Its moment matching approach

provides more informative gradient signals and better estimates of the first two moments of the Bellman target compared to sampling-based approaches, which suffer from high estimator variance caused by Monte Carlo sampling.

*(ii) MOMBO delivers a competitive final training performance.* It ranks highest across all tasks and outperforms other model-based offline reinforcement learning approaches, including COMBO (Yu et al., 2021) with an average normalized reward of 66.8, TT (Janner et al., 2021) with 62.3, and RAMBO (Rigter et al., 2022) with 67.7, all evaluated on the same twelve tasks in the D4RL dataset. We took these scores from Sun et al. (2023).

*(iii) MOMBO delivers superior final training performance when expert data is limited.* MOMBO outperforms the baselines (at $1\%$ in the mixed dataset) in both AULC and normalized reward. For normalized reward, MOMBO achieves 37.0, compared to 32.4 for MOBILE and 27.9 for MOPO, averaged across all tasks at $1\%$. In terms of AULC, MOMBO attains 29.5, outperforming the 28.0 of MOBILE and the 23.5 of MOPO. See Table 3 for details.

*(iv) MOMBO provides more precise estimates of Bellman target uncertainty.* It outperforms both baselines in accuracy for 9 out of 12 tasks and in tightness all 12 tasks in the D4RL dataset.

## 6 Conclusion

The main objective of MOMBO is to accurately quantify the uncertainty surrounding a Bellman target estimate caused by the error in predicting the next state. We achieve this by deterministically propagating uncertainties through the value function with moment matching and introducing pessimism by constructing a lower confidence bound on the target Q-values. We analyze our model theoretically and evaluate its performance in various environments. Our findings may lay the groundwork for further theoretical analysis of model-based reinforcement learning algorithms in continuous state-action spaces. Our algorithmic contributions can also be instrumental in both model-based online and model-free offline reinforcement learning setups (An et al., 2021; Bai et al., 2022).

**Limitations.** The accuracy of the learned environment models sets a bottleneck on the performance of MOMBO. The choice of the confidence radius $\beta$ is another decisive factor in model performance. MOMBO shares these weaknesses with the other state-of-the-art model-based offline reinforcement learning methods (Yu et al., 2020; Lu et al., 2021; Sun et al., 2023) and does not contribute to their mitigation. Furthermore, our theoretical analyses rely on the assumption of normally distributed Q-values, which may be improved with heavy-tailed assumed densities. Considering the bounds, a limitation is that we claim a tighter bound compared to a baseline bound that we derived ourselves. However, the most important factor in that bound, $R_{\max}^2/(1-\gamma)^2$, arises from Hoeffding's inequality, i.e., a classical statement. As such, we assume our bound to be rigorous and not inadvertently biased. Finally, our moment matching method is limited to activation functions for which the first two moments are either analytically tractable or can be approximated with sufficient accuracy, which includes all commonly used activation functions. Extensions to transformations such as, e.g., BatchNorm (Ioffe and Szegedy, 2015), or other activation functions remove the analytical tractability for moment matching. However, these are usually not used in our specific applications.

## Acknowledgments and Disclosure of Funding

AA and MH thank the Carlsberg Foundation for supporting their research under the grant number CF21-0250. We are grateful to Birgit Debrabant for her valuable inputs in the development of some theoretical aspects. We also thank all reviewers and the area chair for improving the quality of our work by their thorough reviews and active discussion during the rebuttal phase.

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

# APPENDIX

## A    Related work

**Offline reinforcement learning.**    The goal of offline reinforcement learning is to derive an optimal policy from fixed datasets collected through interactions with a behavior policy in the environment. This approach is particularly relevant in scenarios where direct interaction with the environment is costly or poses potential risks. The applications of offline reinforcement learning span various domains, including robotics (Singh et al., 2020; Micheli et al., 2022) and healthcare (Shortreed et al., 2011; Nie et al., 2021). Notably, applying off-policy reinforcement learning algorithms directly to offline reinforcement learning settings often fails due to issues such as overestimation bias and distributional shift. Research in offline reinforcement learning has evolved along two main avenues: model-free and model-based approaches.

**Model-based offline reinforcement learning.**    Dyna-style model-based reinforcement learning (Sutton, 1990; Janner et al., 2019) algorithms employ a environment model to simulate the true transition model, generating a synthetic dataset that enhances sample efficiency and serves as a surrogate environment for interaction. Model-based offline reinforcement learning methods vary in their approaches to applying conservatism using the environment model. For instance, MOPO (Yu et al., 2020) and MORel (Kidambi et al., 2020) learn a pessimistic value function by penalizing the MDP generated by the environment model, with rewards being penalized based on different uncertainty measures of the environment model. COMBO (Yu et al., 2021) is a Dyna-style model-based approach derived from conservative Q-learning (Kumar et al., 2020). RAMBO (Rigter et al., 2022) introduces conservatism by adversarially training the environment model to minimize the value function, thus ensuring accurate predictions. MOBILE (Sun et al., 2023) penalizes the value function based on the uncertainty of the Bellman operator through sampling. CBOP (Jeong et al., 2023) introduces conservatism by applying a lower confidence bound to the value function with an adaptive weighting of $h$-step returns in the model-based value expansion framework (Feinberg et al., 2018).

**Uncertainty-driven offline reinforcement learning**    approaches apply penalties in various ways. As model-free methods, EDAC (An et al., 2021) applies penalties based on ensemble similarities, while PBRL (Bai et al., 2022) penalizes based on the disagreement among bootstrapped Q-functions. Uncertainty-driven model-based offline reinforcement learning algorithms adhere to the meta-algorithm known as pessimistic value iteration (Jin et al., 2021). This meta-algorithm introduces pessimism through a $\xi$-uncertainty quantifier to minimize the Bellman approximation error. MOPO (Yu et al., 2020) and MOBILE (Sun et al., 2023) can be seen as practical implementations of this meta-algorithm. MOPO leverages prediction uncertainty from the environment model in various ways, as explored by Lu et al. (2021), while MOBILE uses uncertainty in the value function for the synthetic dataset through sampling. MOMBO is also based on PEVI, and our $\xi$-uncertainty quantifier is derived from the uncertainty of the Bellman target value, which does not rely on sampling. Instead, it directly propagates uncertainty from the environment models to the value function, thanks to moment matching.

**Model-free offline reinforcement learning**    algorithms can be broadly categorized into two groups: policy regularization and value regularization. Policy regularization methods aim to constrain the learned policy to prevent deviations from the behavior policy (Fujimoto et al., 2019; Kumar et al., 2019; Wu et al., 2019b; Peng et al., 2019; Siegel et al., 2020). For instance, during training, TD3-BC (Fujimoto and Gu, 2021) incorporates a behavior cloning term to regulate its policy. On the other hand, value regularization methods (Kostrikov et al., 2021; Xie et al., 2021; Cheng et al., 2022) introduce conservatism during the value optimization phase using various approaches. For example, CQL (Kumar et al., 2020) penalizes Q-values associated with out-of-distribution actions to avoid overestimation. Similarly, EDAC (An et al., 2021), which employs ensemble value networks, applies a similar penalization strategy based on uncertainty measures over Q-values.

**Uncertainty propagation**    via various moment matching-based approaches has been well-researched, primarily in the area of Bayesian deep learning (Frey and Hinton, 1999; Wang and

Manning, 2013; Wu et al., 2019a), with applications, e.g., in active learning (Haußmann et al., 2019) and computer vision (Gast and Roth, 2018).

## B    Theoretical analysis

This section contains proofs for all the theoretical statements made in the main text.

### B.1    Prerequisites

We first restate and extend the following result on moment matching with the ReLU activation function. Our focus on the ReLU activation throughout this work is due to the fact that it is currently the most commonly used activation function in deep reinforcement learning. One could derive similar results for any piecewise linear activation function and several others as well.

**Lemma 1**    *(Moment matching). For $X \sim \mathcal{N}(X|\mu, \sigma^2)$ and $Y = \max(0, X)$, we have*

$\quad$ *(i)* $\quad \widetilde{\mu} \triangleq \mathbb{E}[Y] = \mu\Phi(\alpha) + \sigma\phi(\alpha), \qquad \widetilde{\sigma}^2 \triangleq \mathrm{var}[Y] = (\mu^2 + \sigma^2)\Phi(\alpha) + \mu\sigma\phi(\alpha) - \widetilde{\mu}^2$

*where $\alpha = \mu/\sigma$, and $\phi(\cdot)$, $\Phi(\cdot)$ are the probability density function (pdf) and cumulative distribution function (cdf) of the standard normal distribution, respectively. Additionally, we have that*

$$(ii) \quad \widetilde{\mu} \geq \mu \quad and \quad (iii) \quad \widetilde{\sigma}^2 \leq \sigma^2.$$

*Proof of Lemma 1.* See Frey and Hinton (1999) for the derivation of *(i)*, i.e., $\widetilde{\mu}$ and $\widetilde{\sigma}^2$.

Statement *(ii)* holds as

$$\widetilde{\mu} = \mathbb{E}[Y] = \int_0^\infty xp(x)dx \geq \int_{-\infty}^0 xp(x)dx + \int_0^\infty xp(x)dx = \mu.$$

Note that

$$\mathbb{E}[Y^2] = \int_0^\infty x^2 p(x)dx \leq \int_{-\infty}^0 x^2 p(x)dx + \int_0^\infty x^2 p(x)dx = \mathbb{E}[X^2].$$

Therefore, we get *(iii)* by

$$\widetilde{\sigma}^2 = \mathrm{var}[Y] = \mathbb{E}[Y^2] - \widetilde{\mu}^2 \leq \mathbb{E}[X^2] - \mu^2 = \sigma^2$$

concluding the proof. $\qquad\qquad\square$

Although we do not require the following lemma in our final results, a helpful result is the following inequality between the cdfs of two normally distributed random variables $X$ and its matched counterpart $\widetilde{X}$.

**Lemma 4** (*An inequality between normal cdfs*). *For two normally distributed random variables $X \sim \mathcal{N}(X|\mu, \sigma^2)$ and $\widetilde{X} \sim \mathcal{N}(\widetilde{X}|\widetilde{\mu}, \widetilde{\sigma}^2)$ with $\widetilde{\mu}\sigma \geq \mu\widetilde{\sigma}$, the following inequality holds:*

$$F_{\widetilde{X}}(u) \leq F_X(u) \qquad for\ u \leq 0.$$

Given $\widetilde{\mu}$ and $\widetilde{\sigma}^2$ as in Lemma 1, the assumptions of this lemma hold and it is applicable to our moment matching propagation.

*Proof of Lemma 4.* Assume $u \leq 0$. We have that

$$F_{\widetilde{X}}(u) \leq F_X(u) \quad \Leftrightarrow \quad \Phi\left(\frac{u - \widetilde{\mu}}{\widetilde{\sigma}}\right) \leq \Phi\left(\frac{u - \mu}{\sigma}\right).$$

As it is a monotonically increasing function, this in turn is equivalent to

$$\frac{u - \widetilde{\mu}}{\widetilde{\sigma}} \leq \frac{u - \mu}{\sigma} \quad \Leftrightarrow u\sigma - \widetilde{\mu}\sigma \leq u\widetilde{\sigma} - \mu\widetilde{\sigma}$$

$$\Leftrightarrow u(\sigma - \widetilde{\sigma}) \leq \widetilde{\mu}\sigma - \mu\widetilde{\sigma}$$

which holds as $u(\sigma - \widetilde{\sigma}) < 0$ via Lemma 1 for $u < 0$, and $\widetilde{\mu}\sigma \geq \mu\widetilde{\sigma}$ by assumption. $\qquad\square$

We provide the following definition, as we derive our bounds using Wasserstein distances.

**Definition 1** (*Wasserstein distance*). Let $(M, d)$ denote a metric space, and let $p \in [1, \infty]$. The *p-Wasserstein distance* between two probability measures $\rho_1$ and $\rho_2$ on $M$, with finite $p$-moments, is defined as

$$W_p(\rho_1, \rho_2) \triangleq \inf_{\gamma \in \chi(\rho_1, \rho_2)} \left( \mathbb{E}_{(x,y) \sim \gamma} [d(x, y)^p] \right)^{1/p}$$

where $\chi(\cdot, \cdot)$ represents the set of all couplings of two measures, $\rho_1$ and $\rho_2$.

For $p = 1$, the 1-Wasserstein distance, $W_1$, can be simplified (see, e.g., Villani et al., 2009) to

$$W_1(\rho_1, \rho_2) = \int_{\mathbb{R}} |F_1(x) - F_2(x)| dx \tag{2}$$

where $\rho_1, \rho_2$ are two probability measures on $\mathbb{R}$, and $F_1(\cdot)$ and $F_2(\cdot)$ are their respective cdfs.

Given this definition, we prove the following two bounds on $W_1$.

**Lemma 5** (*Wasserstein inequalities*). *The following inequalities hold for the 1-Wasserstein metric*

*(i)* $W_1(\rho_U^1, \rho_U^2) \leq \|A\| W_1(\rho_V^1, \rho_V^2)$, *for* $U = AV + b$.

*(ii)* $W_1(\rho_Z^1, \rho_Z^2) \leq W_1(\rho_U^1, \rho_U^2)$, *for* $Z = g(U)$ *with* $g(\cdot)$ *locally K-Lipschitz with* $K \leq 1$

*where $\rho_x$ is the density of $x$, $U \in \mathbb{R}^{d_u}$, $V \in \mathbb{R}^{d_v}$, $A \in \mathbb{R}^{d_u \times d_v}$, and $b \in \mathbb{R}^{d_u}$.*

*Proof of Lemma 5.* Given the Kantorovich-Rubinstein duality (Villani et al., 2009) for $W_1$ we have

$$W_1(\mu, \nu) = \frac{1}{K} \sup_{\text{lip}(f) \leq K} \mathbb{E}_{x \sim \mu} [f(x)] - \mathbb{E}_{y \sim \nu} [f(y)]$$

for any two probability measures $\mu$, $\nu$, and where $\text{lip}(f) < K$ specifies the family of $K$-Lipschitz functions.

We use this to first prove *(ii)*. For $Z = g(U)$ we have that

$$\begin{aligned}
W_1(\rho_Z^1, \rho_Z^2) &= \sup_{\text{lip}(f) \leq 1} \mathbb{E}_{z \sim \rho_Z^1} [f(z)] - \mathbb{E}_{z \sim \rho_Z^2} [f(z)] \\
&= \sup_{\text{lip}(f) \leq 1} \mathbb{E}_{u \sim \rho_U^1} [f(g(u))] - \mathbb{E}_{u \sim \rho_U^2} [f(g(u))], \qquad \text{change of variables} \\
&\leq \sup_{\text{lip}(h) \leq 1} \mathbb{E}_{u \sim \rho_U^1} [h(u)] - \mathbb{E}_{u \sim \rho_U^2} [h(u)] = W_1(\rho_U^1, \rho_U^2)
\end{aligned}$$

where the inequality holds as $f \circ g$ has a Lipschitz constant $L \leq 1$, i.e., we end up with a supremum over a smaller class of Lipschitz functions which we revert in the inequality. This gives us the desired inequality.

To prove *(i)* we use that for $\|A\| \leq 1$, the transformation $g(V) \triangleq 1$ is 1-Lipschitz and the result follows via *(ii)*. For $\|A\| > 1$ we use that $g(\cdot)$ is $K$-Lipschitz and the result follows by adapting the proof accordingly. $\square$

Finally, the following lemma allows us to relate the 1-Wasserstein distance of two transformed distributions to an absolute difference between the respective transforming functions.

**Lemma 6** (*Wasserstein to suboptimality*). *Consider two probability distributions $P_x, P_u$ defined on a measurable space $(\mathcal{S}, \sigma(\mathcal{S}))$ and two transformed random variables $y = f(x) \sim P_y$ (with $x \sim P_x$), and $z = g(u) \sim P_z$ (with $u \sim P_u$) for functions $f, g : \mathcal{S} \to \mathbb{R}$. If*

$$W_1(P_y, P_z) \leq C \qquad then \qquad |\mathbb{E}_{x \sim P_x} [f(x)] - \mathbb{E}_{u \sim P_u} [g(u)]| \leq C$$

*for any constant $C \geq 0$.*

*Proof of Lemma 6.* Assume that $W_1(P_y, P_z) \leq C$ for some $C \geq 0$. The Kantorovich-Rubinstein duality (Villani et al., 2009), with $K = 1$ gives us

$$W_1(P_y, P_z) = \sup_{\text{lip}(h) \leq 1} \mathbb{E}_{y \sim P_y} [h(y)] - \mathbb{E}_{z \sim P_z} [h(z)] \leq C.$$

As the identity function $\lambda : x \mapsto x$ is 1-Lipschitz, we can lower bound the supremum further with

$$\mathbb{E}_{y \sim P_y} [y] - \mathbb{E}_{z \sim P_z} [z] \leq \sup_{\text{lip}(h) \leq 1} \mathbb{E}_{y \sim P_y} [h(y)] - \mathbb{E}_{z \sim P_z} [h(z)] \leq C.$$

Using what is colloquially known as the law of the unconscious statistician (Casella and Berger, 2024) , we have that

$$\mathbb{E}_{y \sim P_y} [y] = \mathbb{E}_{x \sim P_x} [f(x)]$$

and analogously for $\mathbb{E}_{z \sim P_z} [z]$. Therefore

$$\mathbb{E}_{y \sim P_y} [y] - \mathbb{E}_{z \sim P_z} [z] = \mathbb{E}_{x \sim P_x} [f(x)] - \mathbb{E}_{u \sim P_u} [g(u)]$$

which gives us

$$\mathbb{E}_{x \sim P_x} [f(x)] - \mathbb{E}_{u \sim P_u} [g(u)] \leq C.$$

As $W_1$ is symmetric, we can follow the same argument and additionally have

$$\mathbb{E}_{u \sim P_u} [g(u)] - \mathbb{E}_{x \sim P_x} [f(x)] \leq C.$$

Using that for $x \in \mathbb{R}$ with $x < C$ and $-x < C$ it follows that $|x| < C$, we can combine these two inequalities and get

$$\left| \mathbb{E}_{x \sim P_x} [f(x)] - \mathbb{E}_{u \sim P_u} [g(u)] \right| \leq C$$

as desired. $\qquad\square$

## B.2 Proof of theorem 2

Proving Theorem 2 requires the following upper $W_1$ bound for an MLP with a 1-Lipschitz activation function.

**Lemma 7** ($W_1$ *bound on an MLP*)**.** *Consider an L-layer MLP $f(\cdot)$*

$$Y_l = A_l^\top X_{l-1} + b_l, \qquad X_l = \mathrm{r}(Y_l), \qquad l = 1, \dots, L$$

*where $A_l$, $b_l$ are the weight matrix and bias vector for layer $l$, respectively, and $\mathrm{r}(\cdot)$ denotes an elementwise 1-Lipschitz activation function. We write $Y \triangleq Y_L = f(X_0)$, where $X_0$ is the input. Assuming two measures $X_1 \sim \rho_X^1$ and $X_2 \sim \rho_X^2$, we have that*

$$W_1(\rho_Y^1, \rho_Y^2) \leq \prod_{l=1}^{L} \|A_l\| W_1(\rho_X^1, \rho_X^2).$$

*Proof of Lemma 7.* We have that

$$
\begin{aligned}
W_1(\rho_Y^1, \rho_Y^2) &\leq \|A_L\| W_1(\rho_{X_{L-1}}^1, \rho_{X_{L-1}}^2), && \text{via Lemma 5 (i)} \\
&= \|A_L\| W_1(\rho_{h(Y_{L-1})}^1, \rho_{h(Y_{L-1})}^2) \leq \|A_L\| W_1(\rho_{Y_{L-1}}^1, \rho_{Y_{L-1}}^2), && \text{via Lemma 5 (ii)} \\
&\leq \cdots \leq \prod_{l=1}^{L} \|A_l\| W_1(\rho_X^1, \rho_X^2).
\end{aligned}
$$

$\square$

The $W_1$ distance between two measures on the output distribution is upper bounded by the $W_1$ distance between the corresponding input measures with a factor given by the product of the norms of the weight matrices $A_l$. This result allows us to provide a probabilistic bound on the $W_1$ distance between a normal $\rho_Y$ and its empirical, sampling-based, estimate $\hat{\rho}_Y$.

The following lemma allows us to extend this result to a probabilistic sampling-based upper bound.

**Lemma 8** (*Sampling-based MLP bound*)**.** *For an L-layer MLP $f(\cdot)$ with 1-Lipschitz activation function, let $Y \triangleq Y_L = f(X)$ where $X \sim \mathcal{N}(X|\mu_X, \sigma_X^2)$, the following bound holds:*

$$\mathbb{P}\left( W_1(\rho_Y, \rho_Y^N) \leq \prod_{l=1}^{L} \|A_l\| \sqrt{-\frac{8 \log(\delta/4) R_{\max}^2}{\lfloor N/2 \rfloor (1-\gamma)^2}} \right) \geq 1 - \delta$$

*where $\delta \in (0, 1)$ and $\rho_{\hat{Y}}^N$ is the empirical estimate given $N$ i.i.d. samples.*

*Proof of Lemma 8.* With Hoeffding's inequality (Wasserman, 2004), we have that

$$\mathbb{P}\left(|\hat{\mu}_N - \mu| \geq \varepsilon\right) \leq 2\exp\left(-2N^2\varepsilon^2 \Big/ \sum_{i=1}^{N}\left(\frac{R_{\max}}{1-\gamma} - 0\right)^2\right) = 2\exp\left(-\frac{2\varepsilon^2 N(1-\gamma)^2}{R_{\max}^2}\right)$$

for the empirical mean $\hat{\mu}_N$ of $Y$. As variances are U-statistics with order $m = 2$ and kernel $h = \frac{1}{2}(x-y)^2$ using Hoeffding (1963); Pitcan (2017), we have for the empirical covariance of $Y$, $\hat{\sigma}_N^2$, that

$$\mathbb{P}(\hat{\sigma}_N^2 - \sigma^2 \geq \varepsilon) \leq \exp\left(-\frac{\varepsilon^2\lfloor N/m\rfloor}{2\|h\|_\infty^2}\right) = \exp\left(-\frac{\varepsilon^2\lfloor N/2\rfloor(1-\gamma)^2}{2R_{\max}^2}\right)$$

$$\Leftrightarrow \quad \mathbb{P}(\hat{\sigma}_N^2 - \sigma^2 \leq \varepsilon) \geq 1 - \exp\left(-\frac{\varepsilon^2\lfloor N/2\rfloor(1-\gamma)^2}{2R_{\max}^2}\right)$$

where $\|\cdot\|_\infty$ denotes $L^\infty$-norm. Applying the inequality $\hat{\sigma}_N \leq \sqrt{\sigma^2 + \varepsilon} \leq \sigma + \sqrt{\varepsilon}$, we get

$$\mathbb{P}(\hat{\sigma}_N \leq \sigma + \varepsilon) \geq 1 - \exp\left(-\frac{\varepsilon^2\lfloor N/2\rfloor(1-\gamma)^2}{2R_{\max}^2}\right)$$

$$\Leftrightarrow \quad \mathbb{P}(\hat{\sigma}_N - \sigma \geq \varepsilon) \leq \exp\left(-\frac{\varepsilon^2\lfloor N/2\rfloor(1-\gamma)^2}{2R_{\max}^2}\right).$$

As the same inequality holds for $\sigma - \hat{\sigma}_N \geq \varepsilon$, a union bound give us that

$$\mathbb{P}(|\sigma - \hat{\sigma}_n| \geq \varepsilon) \leq 2\exp\left(-\frac{\varepsilon^2\lfloor N/2\rfloor(1-\gamma)^2}{2R_{\max}^2}\right).$$

Using an analytical upper bound (Chhachhi and Teng, 2023) on the 1-Wasserstein distance between two normal distributions,

$$W_1(\mathcal{N}(\mu_1, \sigma_1^2), \mathcal{N}(\mu_2, \sigma_2^2)) \leq |\mu_1 - \mu_2| + |\sigma_1 - \sigma_2|$$

we have that

$$\begin{aligned}
\mathbb{P}\big(W_1(\mathcal{N}(\mu,\sigma^2), \mathcal{N}(\hat{\mu}_N, \hat{\sigma}_N)) &\geq 2\varepsilon\big) \\
&\leq \mathbb{P}\left(|\mu - \hat{\mu}_N| + |\sigma - \hat{\sigma}_N| \geq 2\varepsilon\right) \\
&\leq \mathbb{P}(|\hat{\mu}_N - \mu| \geq \varepsilon \text{ or } |\hat{\sigma}_N - \sigma| \geq \varepsilon) \\
&\leq \mathbb{P}(|\hat{\mu}_N - \mu| \geq \varepsilon) + \mathbb{P}(|\hat{\sigma}_N - \sigma| \geq \varepsilon), \qquad \text{union bound} \\
&\leq 2\exp(-2Nc\varepsilon^2) + 2\exp(-0.5\lfloor N/2\rfloor c\varepsilon^2) \\
&\leq 4\exp(-0.5\lfloor N/2\rfloor c\varepsilon^2) \triangleq \delta,
\end{aligned}$$

where we use the shorthand notation $c = (1-\gamma)^2/R_{\max}^2$.

Solving for $\bar{\varepsilon} = 2\varepsilon$ gives us

$$\bar{\varepsilon} = \sqrt{-\frac{8\log(\delta/4)}{\lfloor N/2\rfloor c}}.$$

The bound is then given as

$$\mathbb{P}\left(W_1\left(\mathcal{N}(\mu, \sigma^2), \mathcal{N}(\hat{\mu}_N, \hat{\sigma}_N^2)\right) \leq \sqrt{-\frac{8\log(\delta/4)}{\lfloor N/2\rfloor c}}\right) \geq 1 - \delta$$

which gives us with Lemma 7, that

$$\mathbb{P}\left(W_1(\rho_Y, \rho_{\hat{Y}}^N) \leq \prod_{l=1}^{L}\|A_l\|\sqrt{-\frac{8\log(\delta/4)R_{\max}^2}{\lfloor N/2\rfloor(1-\gamma)^2}}\right) \geq 1 - \delta.$$

$\square$

Finally, applying this result to the relation in Lemma 6 allows us to prove the desired non-asymptotic bound on the performance of the sampling-based model.

**Theorem 2** *(Suboptimality of sampling-based PEVI algorithms). For any policy $\pi_{MC}$ learned by $\mathbb{A}_{PEVI}(\widehat{\mathbb{B}}_\pi^\Gamma Q, \widehat{\mathrm{P}})$ using $N$ Monte Carlo samples to approximate the Bellman target with respect to an action-value network defined as an $L$-layer MLP with $1$-Lipschitz activation functions, the following inequality holds for any error tolerance $\delta \in (0,1)$ with probability at least $1 - \delta$*

$$\mathrm{SubOpt}(\pi_{MC}) \le 2H \prod_{l=1}^{L} \|A_l\| \sqrt{-\frac{8\log(\delta/4)R_{\max}^2}{\lfloor N/2 \rfloor (1-\gamma)^2}}.$$

*Proof of Theorem 2.* Define $C = \prod_{l=1}^{L} \|A_l\| \sqrt{-\frac{8\log(\delta/4)R_{\max}^2}{\lfloor N/2 \rfloor (1-\gamma)^2}}$. Then, combining the bound from Lemma 8 with the result from Lemma 6, we get that with probability $1 - \delta$

$$\left| \mathbb{E}_{X \sim \mathcal{N}(\mu_X, \sigma_X^2)} \left[ \widehat{\mathbb{B}}_\pi Q(X) \right] - \mathbb{E}_{X \sim \mathcal{N}(\widehat{\mu}_X, \widehat{\sigma}_X^2)} \left[ \widehat{\mathbb{B}}_\pi Q(X) \right] \right| \le C$$

$$\left| \mathbb{B}_\pi Q(s, a) - \mathbb{E}_{X \sim \mathcal{N}(\widehat{\mu}_X, \widehat{\sigma}_X^2)} \left[ \widehat{\mathbb{B}}_\pi Q(X) \right] \right| \le C$$

$$\left| \mathbb{B}_\pi Q(s, a) - \widehat{\mathbb{B}}_\pi Q(s, a, s') \right| \le C.$$

Note that $X = (s, a, s')$, i.e., the state, action, and next state tuple, is distributed as described in the main text. Therefore $C$ is a $\xi$-uncertainty quantifier, with $\xi = \delta$. Applying Theorem 1 finalizes the proof. $\qquad\square$

## B.3 Proof of theorem 3

In order to prove Theorem 3, i.e., an upper bound on our proposed moment matching approach, we require the following two lemmata.

The first lemma allows us to upper bound the 1-Wasserstein distance between an input distribution and its output distribution after transformation by the ReLU activation function.

**Lemma 2** *(Moment matching bound). For the following three random variables*

$$X \sim \rho_X, \qquad Y = \max(0, X), \qquad \widetilde{X} \sim \mathcal{N}(\widetilde{X}|\widetilde{\mu}, \widetilde{\sigma}^2)$$

*with $\widetilde{\mu} = \mathbb{E}[Y]$ and $\widetilde{\sigma}^2 = \mathrm{var}[Y]$ the following inequality holds*

$$W_1(\rho_Y, \rho_{\widetilde{X}}) \le \int_{-\infty}^{0} F_{\widetilde{X}}(u)du + W_1(\rho_X, \rho_{\widetilde{X}})$$

*where $F_{\widetilde{X}}(\cdot)$ is cdf of $\widetilde{X}$. If $\rho_X = \mathcal{N}(X|\mu, \sigma^2)$, it can be further simplified to*

$$W_1(\rho_Y, \rho_{\widetilde{X}}) \le \widetilde{\sigma}\phi\left(\frac{\widetilde{\mu}}{\widetilde{\sigma}}\right) - \widetilde{\mu}\Phi\left(-\frac{\widetilde{\mu}}{\widetilde{\sigma}}\right) + |\mu - \widetilde{\mu}| + |\sigma - \widetilde{\sigma}|.$$

*Proof of Lemma 2.* For a generic $X \sim \rho_X$, $Y = \max(0, X)$ and $\widetilde{X} \sim \mathcal{N}(\widetilde{\mu}, \widetilde{\sigma}^2)$, we have with $F_Y(u) = \mathbb{1}_{u \ge 0} F_X(u)^2$ and (2) that

$$W_1(\rho_Y, \rho_{\widetilde{X}}) = \int_{\mathbb{R}} \left| F_Y(u) - F_{\widetilde{X}}(u) \right| du$$

$$= \int_{-\infty}^{0} F_{\widetilde{X}}(u)du + \int_{0}^{\infty} \left| F_X(u) - F_{\widetilde{X}}(u) \right| du$$

$$\le \int_{-\infty}^{0} F_{\widetilde{X}}(u)du + \int_{0}^{\infty} \left| F_X(u) - F_{\widetilde{X}}(u) \right| du + \int_{-\infty}^{0} \left| F_X(u) - F_{\widetilde{X}}(u) \right| du$$

---

[2] The indicator function $\mathbb{1}$ is defined as $\mathbb{1}_S = 1$ if the statement $S$ is true and 0 otherwise.

$$= \int_{-\infty}^{0} F_{\widetilde{X}}(u)du + W_1(\rho_X, \rho_{\widetilde{X}}).$$

If $\rho_X = \mathcal{N}(X|\mu, \sigma)$, we can upper bound the $W_1$ distance via Chhachhi and Teng (2023)

$$W_1(\mathcal{N}(\mu_1, \sigma_1^2), \mathcal{N}(\mu_2, \sigma_2^2)) \le |\mu_1 - \mu_2| + |\sigma_1 - \sigma_2|$$

and evaluate the integral

$$\int_{-\infty}^{0} F_{\widetilde{X}}(u)du = \int_{-\infty}^{0} \Phi\left(\frac{u - \widetilde{\mu}}{\widetilde{\sigma}}\right) du = \widetilde{\sigma}\phi\left(\frac{\widetilde{\mu}}{\widetilde{\sigma}}\right) - \widetilde{\mu}\Phi\left(-\frac{\widetilde{\mu}}{\widetilde{\sigma}}\right)$$

where we used that

$$\int \Phi(a + bx)dx \stackrel{c}{=} \frac{1}{b}\big((a + bx)\Phi(a + bx) + \phi(a + bx)\big)$$

where $\stackrel{c}{=}$ signifies equality up to an additive constant. Together this gives us that

$$W_1(\rho_Y, \rho_{\tilde{x}}) \le \sigma\phi\left(\frac{\widetilde{\mu}}{\widetilde{\sigma}}\right) - \widetilde{\mu}\Phi\left(-\frac{\widetilde{\mu}}{\widetilde{\sigma}}\right) + |\mu - \widetilde{\mu}| + |\sigma - \widetilde{\sigma}|.$$

$\square$

**Lemma 3** *(Moment matching MLP bound). Let $f(X)$ be an L-layer MLP with ReLU activation $r(x) = \max(0, x)$. For $l = 1, \dots, L - 1$, the sampling-based forward-pass is*

$$Y_0 = X_s, \qquad Y_l = r(f_l(Y_{l-1})), \qquad Y_L = f_L(Y_{L-1})$$

*where $f_l(\cdot)$ is the l-th layer and $X_s$ a sample of $\mathcal{N}(X|\mu_X, \sigma_X^2)$. Its moment matching pendant is*

$$\widetilde{X}_0 \sim \mathcal{N}(\widetilde{X}_0|\mu_X, \sigma_X^2), \qquad \widetilde{X}_l \sim \mathcal{N}\left(\widetilde{X}_l \Big| \mathbb{E}\left[r(f_l(\widetilde{X}_{l-1}))\right], \text{var}\left[r(f_l(\widetilde{X}_{l-1}))\right]\right).$$

*The following inequality holds for $\widetilde{\rho}_Y = \rho_{\widetilde{X}_L} = \mathcal{N}(\widetilde{X}_L|\mathbb{E}[f(\widetilde{X}_{L-1})], \text{var}[f(\widetilde{X}_{L-1})])$.*

$$W_1(\rho_Y, \widetilde{\rho}_Y) \le \sum_{l=2}^{L} \left(G(\widetilde{X}_{l-1}) + C_{l-1}\right) \prod_{j=l}^{L} \|A_j\|$$

*with* $\quad G(\widetilde{X}_l) = \widetilde{\sigma}_l\phi\left(\frac{\widetilde{\mu}_l}{\widetilde{\sigma}_l}\right) - \widetilde{\mu}_l\Phi\left(-\frac{\widetilde{\mu}_l}{\widetilde{\sigma}_l}\right) \le 1, \qquad C_l \le |A_l\widetilde{\mu}_{l-1} - \widetilde{\mu}_l| + \left|\sqrt{A_l^2\widetilde{\sigma}_{l-1}^2} - \widetilde{\sigma}_l\right|$

*where $\sqrt{\cdot}$ and $(\cdot)^2$ are applied elementwise.*

*Proof of Lemma 3.* Using Lemma 2 we have that the $W_1$ distance between the deterministic forward pass $\rho_{Y_l}$ and the matching-based forward pass $\rho_{\widetilde{X}_l}$ is given as

$$W_1(\rho_{Y_l}, \rho_{\widetilde{X}_l}) \le G(\widetilde{X}_l) + W_1(\rho_{f_l(Y_{l-1})}, \rho_{\widetilde{X}_l}), \qquad \text{where } G(X) \triangleq \int_{-\infty}^{0} F_X(u)du$$

$$\le G(\widetilde{X}_l) + W_1(\rho_{f_l(Y_{l-1})}, \rho_{f_l(\widetilde{X}_{l-1})})$$
$$\quad + W_1(\rho_{f_l(\widetilde{X}_{l-1})}, \rho_{\widetilde{X}_l}), \qquad\qquad \text{via the triangle inequality}$$
$$\le G(\widetilde{X}_l) + \|A_l\|W_1(\rho_{Y_{l-1}}, \rho_{\widetilde{X}_{l-1}}) + C_l, \qquad\qquad C_l \triangleq W_1(\rho_{f_l(\widetilde{X}_{l-1})}, \rho_{\widetilde{X}_l})$$
$$\le G(\widetilde{X}_l) + \|A_l\|\Big(G(\widetilde{X}_{l-1})$$
$$\quad + \|A_{l-1}\|W_1(\rho_{Y_{l-2}}, \rho_{\widetilde{X}_{l-2}}) + C_{l-1}\Big) + C_l.$$

That is, for the entire MLP we have

$$W_1(\rho_Y, \widetilde{\rho}_Y) = W_1(\rho_{Y_L}, \rho_{\widetilde{X}_L}) = \|A_L\|W_1(\rho_{Y_{L-1}}, \rho_{\widetilde{X}_{L-1}})$$
$$\le \|A_L\|\left(G(\widetilde{X}_{L-1}) + \|A_{L-1}\|W_1(\rho_{Y_{L-2}}, \rho_{\widetilde{X}_{L-2}}) + C_{L-1}\right)$$

$$\leq \sum_{l=3}^{L} \left( G(\widetilde{X}_{l-1}) + C_{l-1} \right) \prod_{j=l}^{L} \|A_j\| + W_1(\rho_{Y_1}, \rho_{\widetilde{X}_1}) \prod_{l=2}^{L} \|A_l\|$$

where, finally,

$$W_1(\rho_{Y_1}, \rho_{\widetilde{X}_1}) \leq \sigma\phi\left(\frac{\widetilde{\mu}}{\widetilde{\sigma}}\right) - \widetilde{\mu}\Phi\left(-\frac{\widetilde{\mu}}{\widetilde{\sigma}}\right) + |A_1\mu_X - \widetilde{\mu}_1| + \left|\sqrt{A_1^2\sigma_X^2} - \widetilde{\sigma}_1\right|.$$

$G(\widetilde{X}_l)$ and $C_l$ are given as

$$G(\widetilde{X}_l) = \widetilde{\sigma}_l\phi\left(\frac{\widetilde{\mu}_l}{\widetilde{\sigma}_l}\right) - \widetilde{\mu}_l\Phi\left(-\frac{\widetilde{\mu}_l}{\widetilde{\sigma}_l}\right), \qquad C_l \leq |A_l\widetilde{\mu}_{l-1} - \widetilde{\mu}_l| + \left|\sqrt{A^2\widetilde{\sigma}_{l-1}^2} - \widetilde{\sigma}_l\right|$$

where $\sqrt{\cdot}$ and $(\cdot)^2$ are applied elementwise. $\qquad\qquad\square$

**Theorem 3** *(Suboptimality of moment matching-based PEVI algorithms). For any policy $\pi_{MM}$ derived with $\mathbb{A}_{PEVI}(\widetilde{\mathbb{B}}_\pi^\Gamma Q, \widehat{P})$ learned by a penalization algorithm that uses moment matching to approximate the Bellman target with respect to an action-value network defined as an L-layer MLP with 1-Lipschitz activation functions, the following inequality holds*

$$SubOpt(\pi_{MM}) \leq 2H \sum_{l=2}^{L} \left( G(\widetilde{X}_{l-1}) + C_{l-1} \right) \prod_{j=l}^{L} \|A_j\|.$$

*Proof of Theorem 3.* The proof is analogous to the one of Theorem 2. Note that $X = (s, a, s')$, i.e., the state-action pair, is distributed as described in the main text.

Define $C = \sum_{l=2}^{L} \left( G(\widetilde{X}_{l-1}) + C_{l-1} \right) \prod_{j=l}^{L} \|A_j\|$. By combining this theorem with the results from Theorem 1, we have that

$$\left| \mathbb{E}_{X\sim\mathcal{N}(\mu_X, \sigma_X^2)} \left[ \widetilde{\mathbb{B}}_\pi Q(X) \right] - \mathbb{E}_{X\sim\mathcal{N}(\widetilde{\mu}_X, \widetilde{\sigma}_X^2)} \left[ \widetilde{\mathbb{B}}_\pi Q(X) \right] \right| \leq C$$

$$\left| \mathbb{B}_\pi Q(s, a) - \mathbb{E}_{X\sim\mathcal{N}(\widetilde{\mu}_X, \widetilde{\sigma}_X^2)} \left[ \widetilde{\mathbb{B}}_\pi Q(X) \right] \right| \leq C$$

$$\left| \mathbb{B}_\pi Q(s, a) - \widetilde{\mathbb{B}}_\pi Q(s, a, s') \right| \leq C$$

holds deterministically. Therefore, $C$ is a $\xi$-uncertainty quantifier for $\xi = \delta = 0$. Applying Theorem 1 finalizes the proof. $\qquad\square$

## C  Further details on experiments

### C.1  Experiment procedures

#### C.1.1  Moment matching versus Monte Carlo sampling experiment

We sample an initial state $s_1$ from the environments. Next, we use the expert policies $\pi^*$, treated as optimal policies, provided[3] in the D4RL datasets (Fu et al., 2020) to sample action $a_1$ from the expert policy. Using the learned environment model $\widehat{P}$, we predict the next state samples $s_2 \sim \widehat{P}(\cdot|s_1, a_1)$. For each sample, we evaluate the next action-value $Q(s_2, \pi^*(s_2))$ using the previously learned critics as the value function. We then evaluate the next action-value using moment matching $Q_{MM}(s_2, \pi^*(s_2))$ and visualize the distributions. For this experiment, we use the learned critics from seed 0. We repeat this process for all tasks in the D4RL dataset and present the results in Figure 3.

---

[3] https://rail.eecs.berkeley.edu/datasets/offline_rl/ope_policies/onnx/

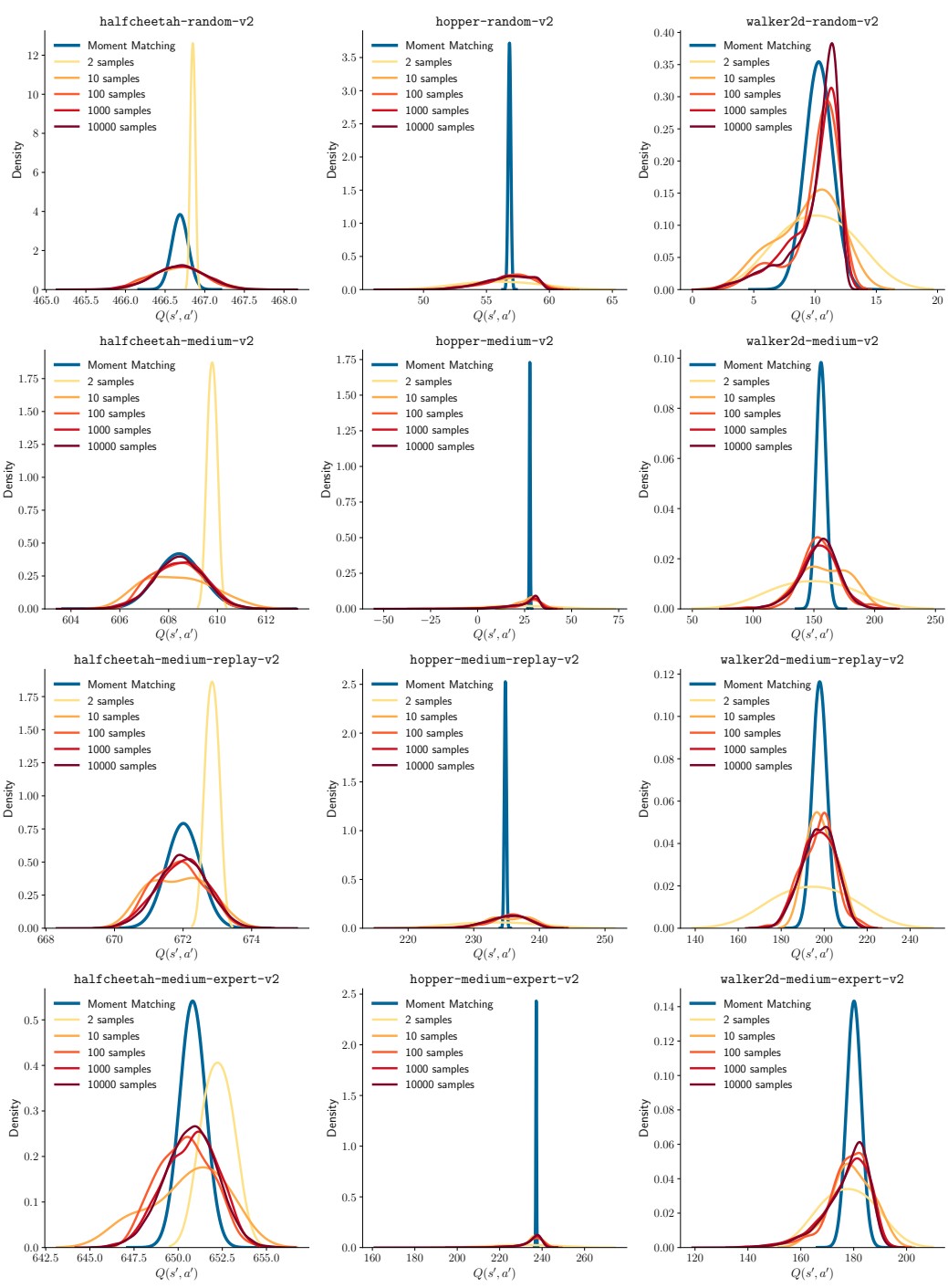

Figure 3: *Moment Matching versus Monte Carlo Sampling.* A comparison of moment matching and Monte Carlo sampling methods for estimating the next value for all tasks in the D4RL dataset.

Table 3: *Performance evaluation on the mixed dataset.* Normalized reward at 2M gradient steps and Area Under the Learning Curve (AULC) (mean±std) scores are averaged across four repetitions. The highest means are highlighted in bold and are underlined if they fall within one standard deviation of the best score. The average normalized score is the average across all tasks, and the average ranking is based on the rank of the mean.

| Dataset Ratio (%) | Dataset Type | Environment | NORMALIZED REWARD (↑) | | | AULC (↑) | | |
|---|---|---|---|---|---|---|---|---|
| | | | MOPO | MOBILE | MOMBO | MOPO | MOBILE | MOMBO |
| 1 | random-medium | halfcheetah | $42.1_{\pm12.0}$ | $51.3_{\pm2.7}$ | $\mathbf{55.4}_{\pm2.1}$ | $41.4_{\pm6.6}$ | $46.0_{\pm2.1}$ | $\mathbf{48.5}_{\pm1.3}$ |
| | | hopper | $\mathbf{48.9}_{\pm25.7}$ | $22.9_{\pm4.0}$ | $\underline{30.4}_{\pm2.0}$ | $\mathbf{35.5}_{\pm3.0}$ | $21.8_{\pm2.2}$ | $27.3_{\pm2.8}$ |
| | | walker2d | $0.1_{\pm0.0}$ | $15.9_{\pm6.3}$ | $\mathbf{22.0}_{\pm0.4}$ | $0.1_{\pm0.0}$ | $13.1_{\pm5.1}$ | $\mathbf{18.7}_{\pm0.7}$ |
| | random-expert | halfcheetah | $33.5_{\pm3.3}$ | $\mathbf{51.8}_{\pm2.0}$ | $41.0_{\pm16.9}$ | $31.2_{\pm1.8}$ | $\mathbf{38.2}_{\pm3.0}$ | $32.3_{\pm8.5}$ |
| | | hopper | $\underline{42.8}_{\pm32.9}$ | $30.7_{\pm16.9}$ | $\mathbf{51.5}_{\pm35.6}$ | $\underline{32.3}_{\pm2.4}$ | $\mathbf{32.7}_{\pm7.0}$ | $\underline{32.0}_{\pm7.2}$ |
| | | walker2d | $-0.1_{\pm0.0}$ | $\underline{21.6}_{\pm0.0}$ | $21.8_{\pm0.4}$ | $0.3_{\pm0.7}$ | $16.6_{\pm0.9}$ | $\mathbf{18.2}_{\pm1.0}$ |
| | | Average on 1 | 27.9 | 32.4 | **37.0** | 23.5 | 28.0 | **29.5** |
| 5 | random-medium | halfcheetah | $\underline{62.1}_{\pm2.7}$ | $\mathbf{62.4}_{\pm1.4}$ | $\underline{62.2}_{\pm3.5}$ | $54.3_{\pm1.5}$ | $56.4_{\pm0.2}$ | $\mathbf{58.1}_{\pm1.1}$ |
| | | hopper | $32.0_{\pm6.7}$ | $\underline{46.8}_{\pm11.8}$ | $\mathbf{60.0}_{\pm27.3}$ | $29.5_{\pm3.7}$ | $38.3_{\pm3.3}$ | $\mathbf{50.1}_{\pm8.2}$ |
| | | walker2d | $5.2_{\pm5.4}$ | $10.6_{\pm6.5}$ | $\mathbf{21.7}_{\pm0.1}$ | $4.7_{\pm4.5}$ | $10.3_{\pm4.7}$ | $\mathbf{18.5}_{\pm0.7}$ |
| | random-expert | halfcheetah | $33.8_{\pm10.2}$ | $\mathbf{54.0}_{\pm22.0}$ | $\underline{39.1}_{\pm19.7}$ | $33.6_{\pm2.3}$ | $\mathbf{49.7}_{\pm10.6}$ | $30.3_{\pm9.7}$ |
| | | hopper | $\mathbf{43.6}_{\pm30.2}$ | $\underline{43.0}_{\pm27.3}$ | $17.1_{\pm12.6}$ | $29.2_{\pm4.8}$ | $\mathbf{42.1}_{\pm4.7}$ | $\underline{39.6}_{\pm3.5}$ |
| | | walker2d | $1.6_{\pm2.0}$ | $13.8_{\pm7.4}$ | $\mathbf{24.0}_{\pm2.6}$ | $2.0_{\pm1.2}$ | $13.3_{\pm3.7}$ | $\mathbf{18.4}_{\pm1.8}$ |
| | | Average on 5 | 29.7 | **38.4** | 37.4 | 25.5 | 35.0 | **35.8** |
| 10 | random-medium | halfcheetah | $66.2_{\pm0.7}$ | $65.4_{\pm0.7}$ | $\mathbf{68.0}_{\pm1.1}$ | $59.5_{\pm1.0}$ | $59.9_{\pm0.4}$ | $\mathbf{61.2}_{\pm1.2}$ |
| | | hopper | $37.3_{\pm9.0}$ | $45.0_{\pm29.0}$ | $\mathbf{58.4}_{\pm14.8}$ | $33.0_{\pm2.1}$ | $\underline{58.4}_{\pm16.8}$ | $\mathbf{60.1}_{\pm8.7}$ |
| | | walker2d | $\mathbf{44.5}_{\pm5.5}$ | $11.1_{\pm6.9}$ | $17.8_{\pm6.7}$ | $\mathbf{24.5}_{\pm4.4}$ | $10.9_{\pm3.8}$ | $16.8_{\pm2.9}$ |
| | random-expert | halfcheetah | $29.5_{\pm9.8}$ | $\mathbf{74.5}_{\pm8.2}$ | $58.1_{\pm16.8}$ | $32.9_{\pm3.8}$ | $\mathbf{64.7}_{\pm2.3}$ | $62.4_{\pm6.6}$ |
| | | hopper | $20.0_{\pm5.7}$ | $\mathbf{80.3}_{\pm21.4}$ | $\underline{77.8}_{\pm29.5}$ | $19.2_{\pm2.1}$ | $40.3_{\pm6.1}$ | $\mathbf{45.5}_{\pm6.2}$ |
| | | walker2d | $4.2_{\pm3.6}$ | $14.9_{\pm7.9}$ | $\mathbf{21.3}_{\pm4.0}$ | $2.9_{\pm1.6}$ | $14.0_{\pm3.4}$ | $\mathbf{17.2}_{\pm1.0}$ |
| | | Average on 10 | 33.6 | 48.5 | **50.2** | 28.7 | 41.3 | **43.9** |
| 50 | random-medium | halfcheetah | $\underline{73.4}_{\pm1.4}$ | $72.0_{\pm0.8}$ | $\mathbf{73.5}_{\pm1.9}$ | $68.4_{\pm0.2}$ | $67.3_{\pm0.8}$ | $\mathbf{69.3}_{\pm0.8}$ |
| | | hopper | $67.2_{\pm34.5}$ | $106.5_{\pm2.0}$ | $\mathbf{108.0}_{\pm0.6}$ | $34.1_{\pm14.1}$ | $\mathbf{80.9}_{\pm5.9}$ | $\underline{82.9}_{\pm4.8}$ |
| | | walker2d | $\mathbf{64.4}_{\pm30.6}$ | $36.1_{\pm25.0}$ | $14.7_{\pm8.2}$ | $\mathbf{58.9}_{\pm16.5}$ | $29.6_{\pm3.2}$ | $37.7_{\pm9.1}$ |
| | random-expert | halfcheetah | $42.1_{\pm10.5}$ | $\mathbf{72.9}_{\pm29.1}$ | $33.4_{\pm12.0}$ | $30.7_{\pm3.6}$ | $\mathbf{47.3}_{\pm6.7}$ | $44.1_{\pm9.6}$ |
| | | hopper | $24.7_{\pm3.7}$ | $\underline{70.3}_{\pm39.8}$ | $\mathbf{88.6}_{\pm37.9}$ | $22.5_{\pm1.8}$ | $62.5_{\pm9.2}$ | $\mathbf{65.4}_{\pm8.5}$ |
| | | walker2d | $\underline{16.0}_{\pm11.4}$ | $15.7_{\pm8.5}$ | $\mathbf{16.4}_{\pm9.7}$ | $7.7_{\pm1.1}$ | $9.8_{\pm4.8}$ | $\mathbf{11.8}_{\pm4.1}$ |
| | | Average on 50 | 48.0 | **62.2** | 55.8 | 37.1 | 49.6 | **51.9** |
| | | Average Score | 34.8 | **45.4** | 45.1 | 28.7 | 38.5 | **40.3** |
| | | Average Ranking | 2.5 | 2.0 | **1.5** | 2.6 | 1.9 | **1.5** |

### C.1.2 Uncertainty quantifier experiment

We generate 10 episode rollouts using the real environment and previously trained policies in evaluation mode. From these rollouts, we select state, action, reward, and next state tuples at every $10^{th}$ step, including the final step. For each selected tuple, we calculate the mean accuracy and tightness scores using the learned environment models. We repeat this process for each of the 4 seeds across every task and report the mean and standard deviation of the accuracies and tightness scores. We estimate the exact Bellman target by taking samples from the policies, and the number of samples is 1000. Table 2 shows the results of this experiment.

### C.1.3 Mixed offline reinforcement learning dataset experiment

The behavior policy of the mixed datasets consists of a combination of `random` and `medium` or `expert` policies, mixed at a specified demonstration ratio. Throughout the experiments, we use the configurations of Cetin et al. (2024). See Appendix C.2 for details regarding the hyperparameters. Results in Table 3 show that MOMBO converges faster with minimal expert input, indicating that it effectively leverages limited information while maintaining competitive performance. Compared to other PEVI approaches, MOMBO demonstrates faster learning, greater robustness, and provides strong asymptotic performance, particularly under the constraints of limited expert knowledge and fixed training budgets, where hyperparameter tuning is not feasible. See Figures 5 to 6 for visualizations of the learning curves.

Table 4: Common hyperparameters and sources used in the experimental pipeline.

| | D4RL | Mixed |
|---|---|---|
| | **Sources** | |
| Dataset | Fu et al. (2020) | Hong et al. (2023) |
| Configuration | Sun et al. (2023) | Hong et al. (2023) |
| Implementation | Sun et al. (2023) | |
| Expert policy | Fu et al. (2020) | — |
| Pretrained environment models | Sun (2023) | — |
| | **Policy learning** | |
| Seeds | $[0, 1, 2, 3]$ | |
| Learning rate actor | 0.0001 | |
| Learning rate critic | 0.0003 | |
| Hidden dimensions actor | $[256, 256]$ | $[1024, 1024, 1024, 1024, 1024]$ |
| Hidden dimensions critic | $[256, 256]$ | $[256, 256, 256],$ |
| Discount factor $(\gamma)$ | 0.99 | |
| Soft update $(\tau)$ | 0.005 | |
| Number of critics | 2 | |
| Batch size | 256 | |
| Learning rate scheduler | Cosine Annealing for Actor | |
| Epoch | 3000 | 2000 |
| Number of steps per epoch | 1000 | |
| Rollout frequency | 1000 steps | |
| Rollout length $(k)$ | See Table 5 | 5 |
| Penalty coefficient $(\beta)$ | See Table 5 | 2.0 |
| Rollout batch size | 50000 | |
| Model retain epochs | 5 | 10 |
| Length of fake buffer $|\widehat{\mathcal{D}}|$ | $5 \times 50000$ | $10 \times 50000$ |
| Real ratio $(p)$ | 0.05 (except `halfcheetah-medium-expert-v2` 0.5) | |
| | **Environment model learning** | |
| Learning rate | 0.001 | |
| Early stop | 5 | |
| Hidden dimensions | $[200, 200, 200, 200]$ | |
| L2 weight decay | $[2.5 \times 10^{-5}, 5 \times 10^{-5}, 7.5 \times 10^{-5}, 7.5 \times 10^{-5}, 1 \times 10^{-4}]$ | |
| Number of ensembles $N_{ens}$ | 7 | |
| Number of elites $N_{elite}$ | 5 | |

## C.2 Hyperparameters and experimental setup

In this section, we provide all the necessary details to reproduce MOMBO. We evaluate MOMBO with four repetitions using the following seeds: $[0, 1, 2, 3]$. We run the experiments using the official repository of MOBILE (Sun et al., 2023),[4] as well as our own model implementation, available at `https://github.com/adinlab/MOMBO`. We list the hyperparameters related to the experimental pipeline in Table 4.

### C.2.1 Environment model training

Following prior works (Yu et al., 2020, 2021; Sun et al., 2023), we model the transition model $\mathrm{P}$ as a neural network ensemble that predicts the next state and reward as a Gaussian distribution. We formulate this as $\widehat{\mathrm{P}}_\theta(s', r) = \mathcal{N}(\mu_\theta(s, a), \Sigma_\theta(s, a))$, where $\mu_\theta$ and $\Sigma_\theta$ are neural networks that model the parameters of the Gaussian distribution. These neural networks are parameterized by $\theta$ and we learn a diagonal covariance $\Sigma_\theta(a, s)$.

---

[4]`https://github.com/yihaosun1124/mobile/tree/4882dce878f0792a337c0a95c27f4abf7e926101`

We use $N_{\text{ens}} = 7$ ensemble elements and select $N_{\text{elite}} = 5$ elite elements from the ensemble based on a validation dataset containing 1000 transitions. Each model in the ensemble consists of a 4-layer feedforward neural network with 200 hidden units, and we apply $L2$ weight decay with the following weights for each layer, starting from the first layer: $[2.5 \times 10^{-5}, 5 \times 10^{-5}, 7.5 \times 10^{-5}, 7.5 \times 10^{-5}, 1 \times 10^{-4}]$. We train the environment model using maximum likelihood estimation with a learning rate of 0.001 and a batch size of 256. We apply early stopping with 5 steps using the validation dataset. The only exception is for `walker2d-medium` dataset, which has a fixed number of 30 learning episodes. We use Adam optimizer (Kingma and Ba, 2015) for the environment model.

We use the pre-trained environment models provided in Sun (2023) to minimize differences and reduce training time. For the mixed dataset experiments, we train environment models using four seeds $(0, 1, 2, 3)$ and use them for each baseline.

### C.2.2 Policy training

**Architecture and optimization details.** We train MOMBO for 3000 episodes on the D4RL dataset and 2000 episodes on the mixed dataset, performing updates to the policy and Q-function 1000 times per episode with a batch size of 256. We set the learning rate for the critics to 0.0003, while the learning rate for the actor is 0.0001. We use the Adam optimizer (Kingma and Ba, 2015) for both the actor and critics. We employ a cosine annealing learning rate scheduler for the actor. For the D4RL dataset, both the actor and critics architectures consist of 2-layer feedforward neural networks with 256 hidden units. For the mixed dataset, the actor uses a 5-layer feedforward neural network with 1024 hidden units, and the critics consist of a 3-layer feedforward neural network with 256 hidden units. Unlike other baselines, our critic network is capable of deterministically propagating uncertainties via moment matching. The critic ensemble consists of two networks. For policy optimization, we mostly follow the SAC (Haarnoja et al., 2018), with a difference in the policy evaluation phase. Using MOBILE's configuration, we apply the deterministic Bellman backup operator instead of the soft Bellman backup operator. We set the discount factor to $\gamma = 0.99$ and the soft update parameter to $\tau = 0.005$. The $\alpha$ parameter is learned during training with the learning rate of 0.0001, except for `hopper-medium` and `hopper-medium-replay`, where $\alpha = 0.2$. We set the target entropy to $-\dim(\mathcal{A})$, where dim is the number of dimensions. We train all networks using a random mixture of the real dataset $\mathcal{D}$ and synthetically generated rollouts $\widehat{\mathcal{D}}$ with real and synthetic data ratios of $p$ and $1 - p$, respectively. We set $p = 0.05$, except for `halfcheetah-medium-expert`, where $p = 0.5$.

**Synthetic dataset generation for policy optimization.** We follow the same procedure for synthetic dataset generation as it is common in the literature. During this phase, we sample a batch of initial states from the real dataset $\mathcal{D}$ and sample the corresponding actions from the current policy. For each state-action pair in the batch, the environment models predict a Gaussian distribution over the next state and reward. We choose one of the elite elements from the ensemble uniformly at random and take a sample from the predicted distribution for the next state and reward. We store these rollout transitions in a synthetic dataset $\widehat{\mathcal{D}}$, where we also keep their variances in the buffer. We generate new rollouts at the beginning of each episode and append them to $\widehat{\mathcal{D}}$. We set the batch size for rollouts to 50000, and store only the rollouts from the last 5 and 10 episodes in $\widehat{\mathcal{D}}$ for the D4RL dataset and the mixed dataset, respectively. For the mixed dataset, the rollout length $(k)$ is 5 for all tasks. Table 5 presents the rollout length for each task for the D4RL dataset.

**Penalty coefficients.** We adopt all configurations from MOBILE with the exception of $\beta$ due to the differences in the scale of the uncertainty estimators. For the D4RL dataset, we provide our choices for $\beta$ in Table 5. For the mixed dataset, we select penalty coefficient $\beta$ as 2.0 for all tasks in order to apply a penalty corresponding to 95% confidence level. We provide all the other shared hyperparameters in Table 4.

**Experiment compute resources.** We perform our experiments on three computational resources: 1) Tesla V100 GPU, Intel(R) Xeon(R) Gold 6230 CPU at 2.10 GHz, and 46 GB of memory; 2) NVIDIA Tesla A100 GPU, AMD EPYC 7F72 CPU at 3.2 GHz, and 256 GB of memory; and 3) GeForce RTX 4090 GPU, Intel(R) Core(TM) i7-14700K CPU at 5.6 GHz, and 96 GB of memory. We measure the computation time for 1000 gradient steps to be approximately 8 seconds for the D4RL dataset and 11 seconds for the mixed dataset on the third device. Assuming the environment models

Table 5: Hyperparameters of `MOMBO` on the D4RL dataset.

| Dataset Type | Environment | Rollout Length $k$ | Penalty Coefficient $\beta$ |
|---|---|---|---|
| random | halfcheetah | 5 | 0.5 |
| | hopper | 5 | 4.5 |
| | walker2d | 5 | 2.5 |
| medium | halfcheetah | 5 | 0.75 |
| | hopper | 5 | 3.0 |
| | walker2d | 5 | 0.75 |
| medium-replay | halfcheetah | 5 | 0.2 |
| | hopper | 5 | 0.15 |
| | walker2d | 1 | 0.5 |
| medium-expert | halfcheetah | 5 | 1.0 |
| | hopper | 5 | 1.5 |
| | walker2d | 1 | 1.5 |

are provided and excluding evaluation time, the total time required to reproduce all MOMBO repetitions is approximately 330 hours for D4RL and 600 hours for the mixed dataset. The computational cost for other experiments is negligible.

## C.3 Visualizations of learning curves

Figures 4 to 6 illustrate the learning curves of PEVI-based approaches, including ours, across gradient steps. In the figures, the thick (dashed/dotted/solid) curve represents the mean normalized rewards across ten evaluation episodes and four random seeds, with the shaded area indicating one standard deviation from the mean. The legend provides the mean and standard deviation of the normalized reward and AULC scores in this order. We show the results for `halfcheetah` in the left panel, `hopper` in the middle panel, and `walker2d` in the right panel. The horizontal axis represents gradient steps and the vertical axis shows normalized episode rewards.

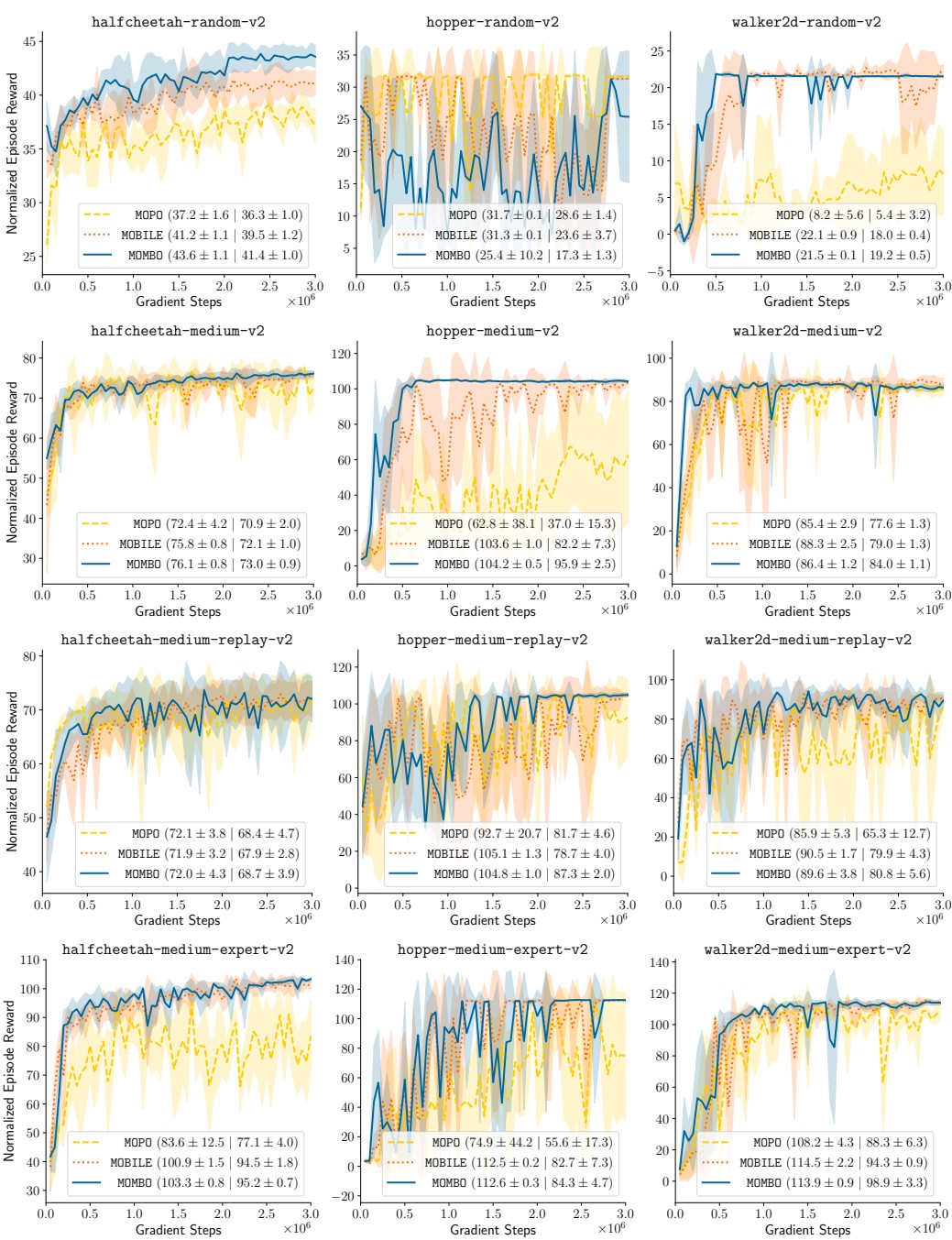

Figure 4: Learning curves for the D4RL dataset.

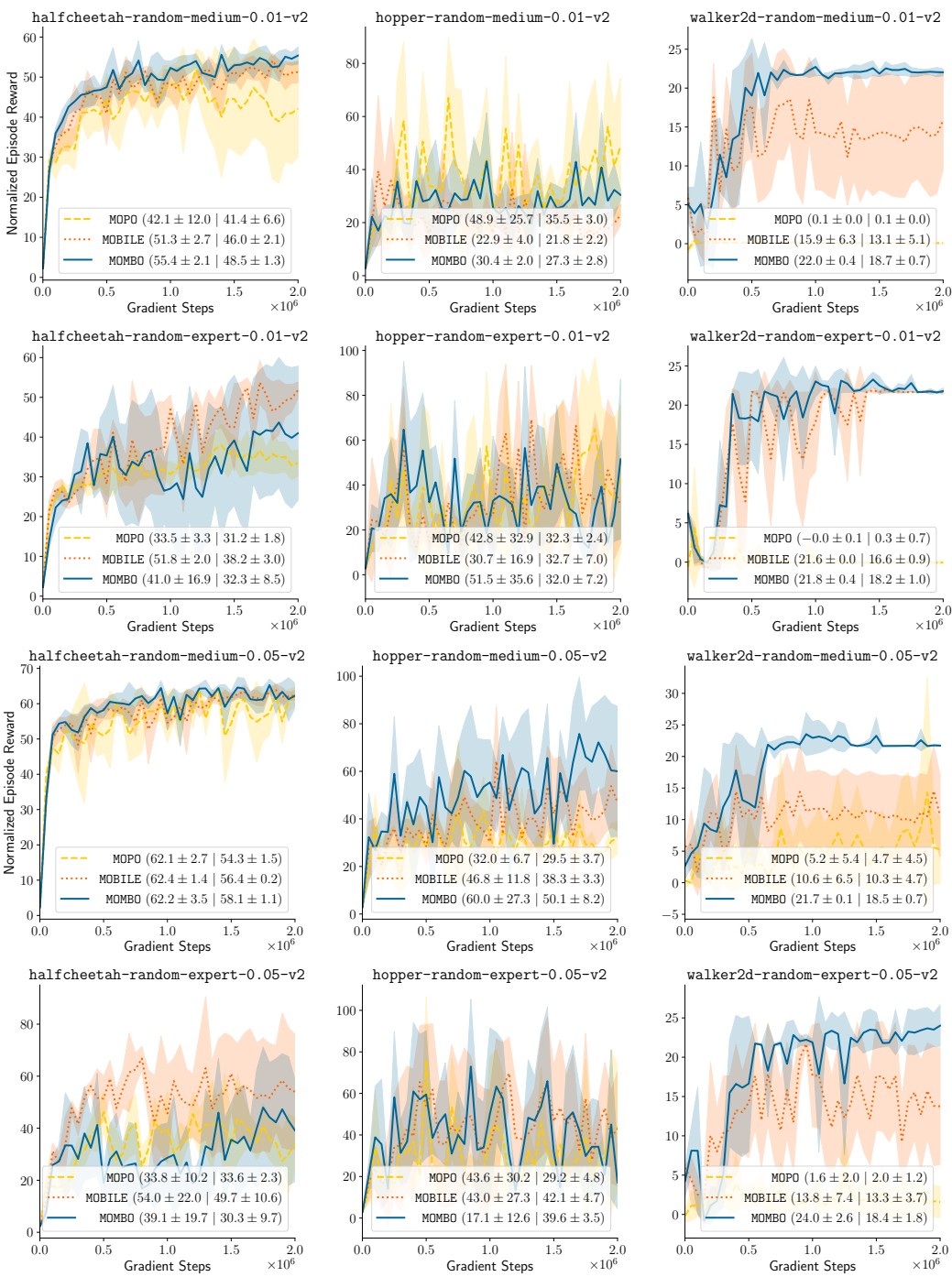

Figure 5: Learning curves for the mixed dataset with trained policy demonstration ratios of `0.01` and `0.05`.

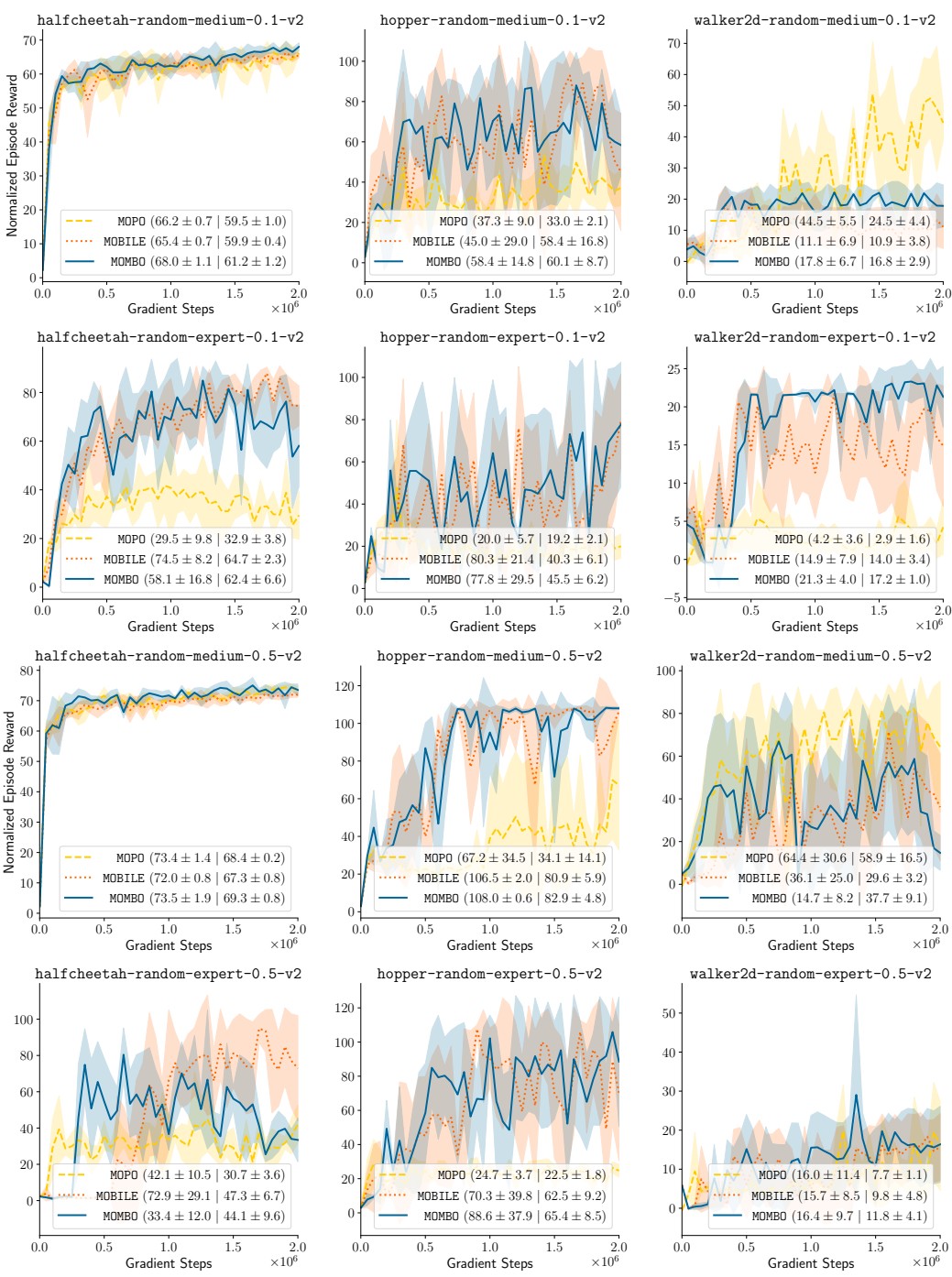

Figure 6: Learning curves for the mixed dataset with trained policy demonstration ratios of `0.1` and `0.5`.

