# OpenReview forum: "Deterministic Uncertainty Propagation for Improved Model-Based Offline Reinforcement Learning"
_NeurIPS.cc/2024/Conference — NeurIPS 2024 poster_

### Official Review · Reviewer_igqb · 2024-07-10

**Soundness:** 2
**Presentation:** 2
**Contribution:** 2
**Rating:** 3
**Confidence:** 4

**Summary:**

To address the high variance issue of model-based offline RL methods, which rely on sampling-based uncertainty estimation, this manuscript proposes MOMBO, a model-based policy optimization method based on moment matching. MOMBO learns a Q-function using moment matching and deterministically propagates uncertainties through the Q-function. Theoretically, using moment matching has tighter upper bounds on value approximation than using Monte Carlo sampling. The authors evaluate MOMBO’s performance across various environments.

**Strengths:**

1. MOMBO is a more stable and sample-efficient approach.

2. There is a tight upper bound on value approximation errors using moment matching compared to Monte Carlo sampling.

**Weaknesses:**

1.There is a clear necessity for the authors to enhance their writing proficiency.

2.The manuscript lists many contributions, which obscures its central thesis. The primary contribution is learning a Q-function using moment matching. However, this approach needs to strike me as sufficiently innovative.

3.How can the alignment of moments from one distribution with those from another ensure the approximation or equivalence of the two?

4.OOD data seems more penalized by high variance, whereas IND data does not generally have high variance. Moreover, there is a need for diversification to improve the variance of Ensemble RL (cf. EDAC, NeurIPS 2021). The moment matching proposed by the authors is similar to transforming uncertainty/variance. Furthermore, the experimental results do not prove the method's validity due to the lack of comparison with the latest SOTA method.

5.Section 3, entitled "Limitations of SOTA model-based offline RL approaches," does not quite capture the issue's essence. Ideally, it should be reframed to reflect a specific shortcoming within a methodology, technique, or theoretical framework. Moreover, the term "SOTA" is inherently time-bound, making it an imprudent choice for a section title, as it may quickly become outdated.

6.The authors follow \mu - \beta \cdot \sigma, where \beta is the weight parameter seems to differ little from most offline RL methods based on uncertainty estimation.

7.The core of the proposed method is based on moment matching; however, the citations to moment matching in the manuscript are too old. Is there any recent work?

8.Only some tests on mujoco environments are insufficient. Isn't it common to perform experimental validation on environments like Antmaze, Adroit, etc.? How do the authors explain this? I suspect that the proposed method will not work in these neglected environments.

**Questions:**

See weaknesses.

**Limitations:**

None.

---

> ### Author Rebuttal · Authors · 2024-08-07
>
> Thank you for this thorough review. We will address your concerns below.
>
> > 1.There is a clear necessity for the authors to enhance their writing proficiency.
>
> Could you be specific as to where the writing proficiency is suboptimal?
>
> > 2.The manuscript lists many contributions, which obscures its central thesis.
>
> Our contribution is three-fold. (i) We propose a novel moment matching and propagation approach for uncertainty-based model-based offline RL.
> (ii) We highlight theoretical differences between it and a sampling-based approach.
> (iii) We provide empirical evidence for the usefulness of our proposal.
> We will restructure the relevant paragraph to make this more clear.
>
> > 3.How can the alignment of moments from one distribution with those from another ensure the approximation or equivalence of the two?
>
> The Wasserstein 1-distance (Def 1) which we consider in this paper measures closeness between two distributions in terms of their cdf not their moments. However, e.g., Wasserstein 2-distance between two univariate Gaussian distributions can be shown to be the sum of the squared difference between their mean and variance terms, i.e., alignment of the moments minimizes their distance based on the $W_2$ metric. Does that answer your question?
>
> > The moment matching proposed by the authors is similar to transforming uncertainty/variance.
>
> Indeed, as we propose a sampling-free approach, this necessitates a way to transform uncertainty as it passes throughout the network. Can the reviewer clarify why this analytical transformation is a weakness and not a feature?
>
> > Furthermore, the experimental results do not prove the method's validity due to the lack of comparison with the latest SOTA method.
>
> EDAC (An et al., 2021) and its improvements, such as SPQR (Lee et al., 2023), focus on model-free learning, whereas we consider model-based algorithms. To our knowledge, MOBILE is the state-of-the-art method for model-based offline learning. Most recent follow-up work such as Luo et al. (2024) and Song et al. (2024) take MOBILE as the base algorithm and complements it with additional features which can be applied also to our method.
> In our general answer, we have included further metrics that highlight the difference between MOBILE's performance and ours.
>
>
>
> > Section 3, entitled "Limitations of SOTA model-based offline RL approaches," does not quite capture the issue's essence. Ideally, it should be reframed to reflect a specific shortcoming within a methodology, technique, or theoretical framework.
>
> Thank you for the proposal. We will change the section title to _Limitations of sampling-based uncertainty propagation_ to highlight the specific limitation we are addressing.
>
> > The authors follow \mu - \beta \cdot \sigma, where \beta is the weight parameter seems to differ little from most offline RL methods based on uncertainty estimation.
>
> We agree, using a lower confidence bound is a standard practice throughout offline RL. That is not a contribution of ours. What is relevant is how this bound is constructed, i.e., in our case $\hat \sigma$.
> See also Theorem 2 in Jin et al. (2021) regarding which derives the desired suboptimality bound on a lower bound estimator.
>
> > The core of the proposed method is based on moment matching; however, the citations to moment matching in the manuscript are too old. Is there any recent work?
>
> There are still minor variations explored in the literature, but these tend to be either only theoretical or too costly, such that there have been no further fundamental new developments. E.g., moment-matching with complete covariance matrices is explored in the current work by Wright et al. (2024). But the cost scales quadratically in the layer width, i.e., becomes intractable for common network sizes. As the authors in that work acknowledge (see their Section 4.3), using the moment matching developed in our references performs just as well or even better. Similarly, Look et al. (2023) explored a way to propagate these covariances in neural stochastic differential equations using Stein's lemma. We will clarify the current state of the literature at the beginning of Section 4.
>
>
> > 8.Only some tests on mujoco environments are insufficient. Isn't it common to perform experimental validation on environments like Antmaze, Adroit, etc.? How do the authors explain this? I suspect that the proposed method will not work in these neglected environments.
>
> We focused on mujoco as a popular and extensive benchmark. See our general answer above for more metrics and a preliminary set of adroit results.
>
>
>
> _____
> An et al., _Uncertainty-Based Offline Reinforcement Learning with Diversified Q-Ensemble_ (NeurIPS 2021)
>
> Jin et al., _Is Pessimism Provably Efficient for Offline RL?_ (ICML 2021)
>
> Lee et al., _SPQR: Controlling Q-ensemble Independence with Spiked Random Model for Reinforcement Learning_ (NeurIPS 2023)
>
> Look et al., _A deterministic approximation to neural SDEs_ (PAMI 2023)
>
> Luo et al., _Reward-Consistent Dynamics Models are Strongly Generalizable for Offline Reinforcement Learning_ (ICLR 2024)
>
> Song et al., _Compositional Conservatism: A Transductive Approach in Offline Reinforcement Learning_ (ICLR 2024)
>
> Wright et al., An Analytic Solution to Covariance Propagation in Neural Networks (AISTATS 2024)

---

> > ### Comment · Reviewer_igqb · 2024-08-12
> >
> > Thanks for the author's response, which has resolved some of my initial concerns. I have thoroughly read the comments from other reviewers and the author's replies, including the general response. Below, I provide explanations for the points raised by the author and list the concerns that are still unclear to me:
> >
> >
> > 1. Improvement Suggestions for Writing:
> >
> > Here are the areas I believe would benefit from refinement in writing:
> >
> > - Sentences are too long and hard to read: Line 20-23 "Applying traditional …"， line 24-27 "Model-based …"， line 171-174 "As the input …"
> >
> > - Conflation of expressions: distributional shift & distribution shift, behavioral policy & behavior policy. Refer to literatures MOBILE and COMBO.
> >
> > - Sentence Construction: On lines 89-104, the frequent use of "when" and consecutive "while" at the beginning of sentences, coupled with a near absence of connecting words, raises concerns about the flow and coherence of the writing.
> >
> > - Passive Voice: Upon reviewing lines 121-125, it has been observed that the text relies heavily on the passive voice. While the passive voice can be appropriate in certain contexts, an overreliance on it can obscure the clarity and directness of the writing.
> >
> > - Writing Conventions: The placement of "E.g.," (Line 146) and "I.e.," (Lines 174, 226) at the beginning of sentences is not in accordance with standard academic writing practices. While these abbreviations are commonly used to provide examples or clarify meanings, their position at the sentence's onset can disrupt the flow and formality expected in scholarly articles.
> >
> > Improvement of Pronoun Usage: The manuscript exhibits multiple instances of ambiguous or incorrect pronoun usage, which can lead to confusion for the reader (as seen on line 135; a comprehensive list is not provided here for the sake of brevity).
> >
> > - Refinement of Expressions for Scholarly Rigor: The manuscript frequently employs expressions that may not meet the standards of precision expected in academic writing, such as "via" (the specific instances are not exhaustively listed here).
> >
> > In summary, the numerous writing issues are a significant shortcoming of this paper. The abovementioned concerns may only encompass some areas that require attention; thus, the authors must invest further effort into meticulously reviewing the entire manuscript.
> >
> >
> > 2. Clarification and Enhancement of Contribution Statement:
> >
> > The conciseness of the stated contributions needs to be improved. The author's response appears somewhat superficial and has not yet captured the essence of the contributions. If I understand correctly, this paper's core contribution lies in employing a moment-matching method to address the variance issues arising from Monte Carlo sampling. However, the authors have not adequately reflected these critical elements, especially considering using the term "bottleneck" as a substitute.
> >
> > 3. Clarification of Research Motivation:
> >
> > The motivation behind the study is somewhat ambiguous. If I have understood correctly, the paper's logic is as follows: Monte Carlo sampling leads to variance issues, which in turn prompts the proposal of a sampling-free method. It seems the authors aim to improve upon Monte Carlo sampling. However, I have concerns:
> >
> > - Is high variance the only issue with ineffective Monte Carlo sampling? Given that the authors have only analyzed the variance issue, it naturally raises the following concern:
> > - Does the variance problem only arise with Monte Carlo sampling?
> > - Compared to other means of addressing the variance issue, what is the motivation for using a sampling-free approach in this paper?
> >
> > In summary, the current manuscript's analysis of motivation is incomplete. I have not grasped the authors' central focus, which is why solving the variance problem necessitates using a sampling-free approach.

---

> > > ### Comment · Reviewer_igqb · 2024-08-12
> > >
> > > 4. Further Experimental Validation and Concerns:
> > >
> > > I appreciate the additional experiments provided by the authors. However, even with the selected tasks, I fail to discern the advantages of MOMBO. The performance of the proposed method does not surpass MOBILE; the only advantage seems to be a slightly smaller standard deviation. However, the high standard deviation in the pen-cloned scenario contradicts the motivation of this paper. Moreover, the authors have corrected the expression from sample efficiency to convergence rate. However, in Figure 1 of the manuscript, it appears that only the halfcheetah-random task shows a faster convergence, with no significant advantage in the convergence rate for the other three tasks when compared to the high performance achieved by MOBILE. In the halfcheetah-medium-replay task, although it starts converging faster, it experiences significant fluctuations later, reflecting instability in the training process. Additionally, the standard deviation in the appendix's Figure 4 for both the halfcheetah-medium-replay and walker2d-medium-replay tasks is also significant.
> > >
> > > In summary, I believe there is room for improvement in this paper's experimental validation, including, but not limited to, a broader comparison of experimental tasks, verification of method effectiveness (for example, modifying linear or ReLU layers to construct ablation studies), and the reliability of statistical results. These concerns also prevent me from confidently assigning a high score.

---

> > > > ### Author Response · Authors · 2024-08-12
> > > > **Clarifications about Experiments and Results**
> > > >
> > > > > 4. Further Experimental Validation and Concerns:
> > > >
> > > > We would like to reiterate our reply to Reviewer BmgH about the interpretation of our key experimental findings.
> > > >
> > > > In a realistic offline RL scenario, it is not possible to measure the evaluation-time performance, hence not trivial to decide how long a model should be trained. Hence, we argue that the steepness of the learning curve is a crucial performance criterion, and the area under the learning curve is an appropriate means to quantify its steepness. As visible in the right-most column of Table 1, MOMBO reaches a significantly steeper learning curve than MOBILE, which is what we would expect from an offline RL algorithm that calculates its reward penalty with less approximation error. The normalized reward results, in turn, serve as a sanity check. They demonstrate that our scheme maintains the original MOBILE performance when trained long enough. Notably, we expect an algorithm with a higher AUC score to deliver better results for a fixed amount of computational time or training data.
> > > >
> > > > We would also like to bring the uncertainty quantification results we reported in the global response to your attention. These results provide direct evidence about why moment matching would address the sampling error more effectively.
> > > >
> > > > > Figure 1 of the manuscript, it appears that only the halfcheetah-random task shows a faster convergence, with no significant advantage in the convergence rate for the other three tasks when compared to the high performance achieved by MOBILE. In the halfcheetah-medium-replay task, although it starts converging faster, it experiences significant fluctuations later, reflecting instability in the training process.
> > > >
> > > > We believe there is a misunderstanding here about the implications of the results we presented. It is true that MOMBO fluctuates after 1 million steps during halfcheetah-medium-replay training a little more than MOBILE. However, this fluctuation is still within the one standard deviation distance of MOBILE's mean performance during the far majority of the training process. The eventual AUC scores of the two models are nearly identical even with similar standard deviations across experiment repetitions: 68.7 ± 3.9 for MOMBO and 67.9 ± 2.8 for MOBILE. We would like to remind that halfcheetah-medium-replay is among the few environments where MOMBO fluctuates more than MOBILE. In the majority of the tasks, MOMBO delivers clearly higher AUC than MOBILE, providing strong evidence of its improved training stability and supporting the central claim of our work.
> > > >
> > > > > Additionally, the standard deviation in the appendix's Figure 4 for both the halfcheetah-medium-replay and walker2d-medium-replay tasks is also significant.
> > > >
> > > > As reported in Table 1, MOMBO delivers smaller standard deviations across experiment repetitions than other models in six of the twelve environments with respect to final reward, while MOBILE does the same for only four. However, one can see MOMBO's advantage over MOBILE more directly from the AUC scores, where it outperforms MOBILE in all environments but one. The legends of Figures 2, 3, and 4 convey the same information as Table 1 for individual environments. You can see in the final and AUC scores presented in the Figure 4 legends that MOMBO's variance across experiment repetitions is comparable to MOBILE, as expected.
> > > >
> > > > > Adroit experiments
> > > >
> > > > We reported new experiments only on the two Adroit environments for which the official GitHub source of the MOBILE paper provides sufficient details regarding the configuration. We did not select the tasks based on other criteria.

---

> > > > > ### Comment · Area_Chair_xd1N · 2024-08-13
> > > > >
> > > > > Hi reviewer igqb,
> > > > >
> > > > > Thanks so much for your lively engagement with the authors so far. We are about to lose access to them in the discussion period (August 13, 11:59pm AoE). I'm hoping that you might be able to weigh in one more time regarding your concerns about the comparison between MOBILE and MOMBO. The authors have made several points in their most recent comment. They argue that the key improvements are learning with fewer training steps and improved uncertainty estimates (see the PDF from the rebuttal); they have expanded upon the halfcheetah results; and they have clarified how the Adroit environments were selected.
> > > > >
> > > > > Could you take a look and let us know the extent to which your concerns have been alleviated by these responses?
> > > > >
> > > > > Thanks!

---

> > > > > ### Comment · Reviewer_igqb · 2024-08-14
> > > > >
> > > > > 1. Could you please clarify why only minor writing changes are considered? If there is no substantial evidence to counter the multiple shortcomings I have highlighted, then my concerns remain unresolved.
> > > > >
> > > > > 2. The authors have yet to adequately address my concerns regarding the motivation behind the research. I kindly request that you respond directly to the three questions I posed in the Clarification of Research Motivation.
> > > > >
> > > > > 3. Based on my understanding, a lower standard deviation typically indicates a faster rate of convergence. However, I am concerned that from Figures 2 through 5, the proposed method demonstrates a fast convergence rate in only a few tasks in 4/12  (Figure 1: HalfCheetah, Figure 2: HalfCheetah, Walker, and Figure 3: Hopper). Given that the convergence rate is not consistently guaranteed, fluctuations are expected, and performance does not exceed that of MOBILE, what is the significance of a low standard deviation? From the current graphs, it appears that MOMBO, despite having a high AUC, does not outperform MOBILE when considering the same number of training steps.

---

> > > > > > ### Author Response · Authors · 2024-08-14
> > > > > > **Further Clarifications**
> > > > > >
> > > > > > 3. “Based on my understanding, a lower standard deviation typically indicates a faster rate of convergence.“
> > > > > >
> > > > > > Can you specify the problem with treating the area under learning curve scores we reported in Table 1 as a quantitative measure of convergence rate and drawing conclusions accordingly? We do understand the merit of looking carefully into individual figures, however, as scientists, we believe more in numbers than bare eye analysis. From the fact that in 11 of the 12 tasks MOMBO delivers higher AUC and similar cross-experiment variance compared to MOBILE, it converges faster.
> > > > > >
> > > > > > MOMBO and MOBILE perform similarly only when both models are trained for 3 million steps. This training time is typically chosen by performing online evaluations. However, in a realistic offline RL setup, we do not have access to the real environment on which we can check whether our model is sufficiently trained. Furthermore, a model that converges faster would fit better to data when its capacity is increased.
> > > > > >
> > > > > > The significance of a low standard deviation is improved training robustness. This comes as a side benefit of eliminating a factor of randomness from the training process and an explanation of why our method reaches faster convergence, which is our main result. MOBILE and MOPO perform comparably only when both are trained long enough. This outcome serves as a consistency check and is not the main point of our contribution.
> > > > > >
> > > > > > We hope we were able to answer your questions. Thanks again for the truly enjoyable scientific discussion we had together. We fully trust your fair scientific judgment and respect any score you give in your final evaluation.

---

> > > ### Author Response · Authors · 2024-08-12
> > > **Clarifications about Contribution Statement and Motivation**
> > >
> > > Thanks for your meticulous review and valuable suggestions. We believe there remains a few points calling for further clarification, but we also fully respect your final decision.
> > >
> > > > 1. Improvement Suggestions for Writing:
> > >
> > > We agree with your points and appreciate your rigor in pointing them out greatly. We promise to handle them in the revised version of our work. However, we kindly express our view that their scope stays within the limits of a minor revision.
> > >
> > > > 2. Clarification and Enhancement of Contribution Statement:
> > > > 3. Clarification of Research Motivation:
> > >
> > > We agree about the need for further improvement here. We believe presenting our core contribution in Section 1, `Our contribution` paragraph as follows would address both points 2 and 3, as well as your concern about the central focus of our work:
> > >
> > > Estimating tight $\xi$-uncertainty quantifiers and using them for reward penalization is a provable recipe for improved offline reinforcement learning (see Theorem 4.2 of Jin et al. (2021)). Reward penalties calculated on the Bellman target are provably tighter $\xi$-uncertainty quantifiers than those calculated on the environment model (see Theorem 3.6 of Sun et al. (2023)). However, the calculation of this quantity necessitates propagating the uncertainty of the estimated next state through a non-linear value function, which is typically modeled by a deep neural net. The commonplace approach is to approximate this quantity by Monte Carlo sampling. Our key novelties include:
> > >
> > > i) detecting the problem of approximation errors on the reward penalty for the first time in the literature
> > > ii) characterizing the problem theoretically by deriving novel concentration inequalities on their commonplace Monte Carlo estimator,
> > > iii) proposing an original solution to this problem by using a moment matching approach originally developed for Bayesian neural net inference for the first time in the context of reward penalty calculation,
> > > iv) quantifying the approximation error incurred by the Monte Carlo estimate on a large set of benchmarks, showing a significant reduction of this error by our proposed moment matching approach, and demonstrating the impact of the error on convergence speed (improved AUC score).
> > >
> > >
> > > -----
> > > Jin et al., _Is Pessimism Provably Efficient for Offline RL?_ (ICML 2021)
> > >
> > > Sun et al., _Model-bellman inconsistency for model-based offline reinforcement learning_ (ICML 2023)

---

> ### Author Response · Authors · 2024-08-14
> **Further Clarifications**
>
> 1. We promise once again to implement all of your suggestions, which comprise:
>     1. Shortening the sentences that are hard to read
>     2. Unifying the allocation of terminology, sticking to their usage in the literature
>     3. Correcting wrongly constructed sentences
>     4. Rephrasing sentences written in passive voice in active voice in the most understandable way
>     5. Correcting improperly used writing conventions throughout the text.
>
> We do think they are extremely important to improve the quality of our work. Their implementation does require a thorough and careful pass over the text, which is a tedious task and we are ambitious to fulfill it following your guidance. We kindly express our opinion that all these corrections are typically handled during the preparation of a camera-ready versions of conference papers. We encounter corresponding period in journal publication processes as **accept with minor revision**.
>
> 2. Here are our answers to your questions:
>
> We would like to first make sure that we agree on the terminology. We study the **variance** of *the Monte Carlo estimators used in the penalty term calculation of model-based offline RL algorithms*. Using an estimator is inevitable while calculating a penalty term based on the Bellman operator. In mathematical terms, there is uncertainty on the next state $s’$ stemming from the error in the learned environment dynamics. This uncertainty needs to be propagated through the state-action value function $Q(s’,a’)$ which appears in the Bellman target calculation. This operation is analytically intractable if $Q$ is a neural network.
>
> > Is high variance the only issue with ineffective Monte Carlo sampling?
>
> We are sure it is not the only issue. Our claim is that it is one of the many important issues as it plays a significant role in training efficiency. We demonstrate why it is the case by developing concentration inequalities for the Monte Carlo estimator and our moment matching based alternative.
>
> > Given that the authors have only analyzed the variance issue, it naturally raises the following concern:” Does the variance problem only arise with Monte Carlo sampling?
>
> As we pose the problem in the first place as the estimator variance of a Monte Carlo estimate, the answer is yes. Reducing estimator variance is major issue in a wide range of problems in statistics. It is handled using methods such as introducing control variates or Rao-Blackwellization. See for instance: Lemieux (2017).
>
>
> The state-of-the-art approach to approximate the uncertainty of the state-action value of the next state is to use Monte Carlo integration, which is represented in our paper with the MOBILE algorithm (Sun et al., 2023). Sampling-free approaches do exist, but only for penalty terms that are computed directly on the estimated error of the model dynamics. We represent this alternative in our paper by the MOPO algorithm (Yu et al., 2021). However, as proven in Theorem 4.3 of the MOBILE paper, this option over-penalizes the reward and causes suboptimal results. The tightness experiment we reported in our global response demonstrates that MOPO applies much higher penalty than MOBILE as this theorem suggests, and our MOMBO applies even less penalty, hence brings about much faster training.
>
> > Compared to other means of addressing the variance issue, what is the motivation for using a sampling-free approach in this paper
>
> There does not exist an alternative way to address this issue yet. Our key novelty is that we are the first to detect its existence and prominence for offline reinforcement learning and provide a solution to mitigate it.
>
>
> -----
> Lemieux, C., _Control Variates_ (Wiley StatsRef: Statistics Reference OnlineL 1-8 2017)
>
> Yu et al., _COMBO: Conservative offline model-based policy optimization_ (NeurIPS 2021)
>
> Sun et al., _Model-bellman inconsistency for model-based offline reinforcement learning_ (ICML 2023)

---

### Official Review · Reviewer_BmgH · 2024-07-11

**Soundness:** 3
**Presentation:** 3
**Contribution:** 3
**Rating:** 6
**Confidence:** 4

**Summary:**

This paper proposes a new uncertainty estimation method for model-based offline reinforcement learning, which uses moment matching to deterministically propagate uncertainties through the Q-function, rather than relying on sampling-based uncertainty estimation. The resulting model, Moment Matching Offline Model-Based Policy Optimization (MOMBO), significantly accelerates training compared to its sampling-based counterparts while maintaining performance.

**Strengths:**

- The paper is clearly written, allowing readers to follow the main arguments.
- It provides a detailed theoretical analysis to demonstrate why moment matching is better than Monte Carlo sampling.

**Weaknesses:**

- The novelty of this method seems relatively weak. It appears to be a minor modification of the uncertainty quantification method used in MOBILE [1]. Additionally, as noted by the authors in the limitations section, this method is also limited to certain activation functions. I want to clarify that I do not think minor modifications lack value or novelty. Instead, the novelty of a method should be evaluated along with its empirical performance. However, the experiments in this paper do not show a strong improvement over the baseline (MOBILE).
- The advantage claimed by the authors is sample efficiency. Firstly, using “sample efficiency” here is incorrect. This criterion is used to evaluate an online RL method. In offline RL, we train an algorithm on a fixed dataset, so “convergence speed” would be more appropriate. Furthermore, this criterion is not critical in an offline setting, where we are more concerned with the asymptotic performance.
- According to PEVI [2], the key factor for an uncertainty-based offline RL method is how tightly it can estimate the Bellman error, which directly determines the performance of the derived policy. The authors do not provide direct evidence (theoretical or empirical) to show that MOMBO can provide a tighter estimation of the Bellman error.

[1] Sun et al. "Model-Bellman Inconsistency for Model-based Offline Reinforcement Learning" (ICML 2023)

[2] Jin et al. "Is Pessimism Provably Efficient for Offline RL?" (ICML 2021)

**Questions:**

To summarize the main points/questions raised in the weaknesses section:

- Could the authors evaluate their method on more challenging tasks to show that it can achieve better asymptotic performance compared to MOBILE?
- Could the authors provide evidence to demonstrate that MOMBO can provide a more correlated/tighter estimation of the ideal uncertainty, i.e., the Bellman error?

I‘m willing to raise my score if all these concerns can be addressed by the authors.

**Limitations:**

The authors have thoroughly discussed the limitations of their method in the paper.

---

> ### Author Rebuttal · Authors · 2024-08-07
>
> Thank you for the detailed review.
>
> > The novelty of this method seems relatively weak. It appears to be a minor modification of the uncertainty quantification method used in MOBILE.
>
> MOPO, MOBILE, and MOMBO are three algorithms derived from the meta-algorithm known as Pessimistic Value Iteration (Jin et al., 2021). They all attempt to quantify uncertainty to learn a pessimistic value function. Therefore, a high degree of similarity among them is expected. However, we demonstrate that our approach surpasses the SOTA (MOBILE) both analytically and empirically.
>
> >  as noted by the authors in the limitations section, this method is also limited to certain activation functions.
>
> Let us clarify that while this limitation exists, it is less restrictive than it sounds. The derivations required to compute the first two moments of a ReLU activation directly generalize to all other piecewise linear activation functions, e.g., Leaky-Relu or PReLU.
> While requiring a more expensive forward pass, the first two moments are also tractable for more modern activation functions such as the ELU and its SELU counterpart.
> To summarize, while the first two moments cannot be computed for all activation functions, they can be for the most relevant and most popular ones covering all mainstream architectures used in critic training in modern RL.
>
>
>
> > The advantage claimed by the authors is sample efficiency. Firstly, using “sample efficiency” here is incorrect. This criterion is used to evaluate an online RL method. In offline RL, we train an algorithm on a fixed dataset, so “convergence speed” would be more appropriate. Furthermore, this criterion is not critical in an offline setting, where we are more concerned with the asymptotic performance.
>
> We agree, that _sample efficiency_ was poorly worded. We adapt the discussion paragraph in the final version of the paper accordingly to make this clear. (Table 2 refers to the new results provided in this rebuttal.)
>
> ```
> Discussion and results. To summarize, our experimental findings are:
> (i) MOMBO matches the state-of-the-art MOBILE in normalized reward. Additionally, it outperforms other model-based offline RL approaches like COMBO (Yu et al., 2021) with 66.8, TT (Janner et al., 2021) with 62.3, and RAMBO (Rigter et al., 2022) with 67.7 as their respective average normalized reward scores across the same twelve tasks. These numbers are cited from Sun et al. (2023).
> (ii) MOMBO has a faster convergence speed. It learns faster and reaches its final performance earlier. This is reflected in its AUC score, where MOMBO outperforms the baselines.
> (iii) MOMBO provides improved uncertainty estimates for the Bellman target. Its moment-matching approach provides it with tighter and more accurate estimates, as shown in Table 2.
> (iv) MOMBO is more robust. It has a lower standard deviation than the normalized reward in six out of twelve tasks. This indicates better stability compared to MOBILE, which has a lower standard deviation in only three tasks. Note that for hopper-random, our model failed to learn in one repetition, leading to a high standard deviation and low average normalized reward. In conclusion, our MOMBO achieves state-of-the-art performance, exhibiting robustness and fast learning capabilities, aligning with the theory.
> ```
>
> > However, the experiments in this paper do not show a strong improvement over the baseline (MOBILE).
>
> Upon three million gradient steps, both MOBILE and MOMBO perform very similar, with a tendency for minor improvements in MOMBO. However, the AUC results, being calculated over ten evaluation results throughout the training process, show a clear distinction in the speed of convergence with MOMBO outperforming MOBILE in all but one environment. Convergence speed is a crucial metric in real applications where it is not feasible to engineer the training duration based on evaluation results, which is the common practice for standard offline RL benchmarks.
> Additionally, this means that for a fixed amount of computational time/training data, we can train more powerful models than MOBILE, which can be expected to then translate into a larger normalized reward as well.
> Finally, the experiments performed for the rebuttal (see general answer) show improvements in accuracy and tightness indicating a better training signal for our proposed approach.
>
>
> > Could the authors evaluate their method on more challenging tasks to show that it can achieve better asymptotic performance compared to MOBILE?
>
>
> We provide additional performance metrics in our general answer demonstrating the improvements our method provides as well as preliminary results on the adroit domain.
>
>
> > Could the authors provide evidence to demonstrate that MOMBO can provide a more correlated/tighter estimation of the ideal uncertainty, i.e., the Bellman error?
>
> We have included an additional experiment in our main response that demonstrates improved performance in terms of accuracy and tightness upon both MOBILE and MOPO.
>
> Please let us know if this answer clarifies your concerns.
>
>
>
>
> -----
> Jin et al., _Is Pessimism Provably Efficient for Offline RL?_ (ICML 2021)
>
> Yu et al., _COMBO: Conservative offline model-based policy optimization_ (NeurIPS 2021)
>
> Janner et al., _Offline reinforcement learning as one big sequence modeling problem_ (NeurIPS 2021)
>
> Rigter et al., _RAMBO-RL: Robust adversarial model-based offline reinforcement learning_ (NeurIPS 2022)
>
> Sun et al., _Model-bellman inconsistency for model-based offline reinforcement learning_ (ICML 2023)

---

> > ### Comment · Reviewer_BmgH · 2024-08-10
> >
> > Thank you for your reply, my concerns have been mostly resolved. I have already improved my score.

---

### Official Review · Reviewer_czcW · 2024-07-11

**Soundness:** 3
**Presentation:** 2
**Contribution:** 3
**Rating:** 5
**Confidence:** 2

**Summary:**

** I am unfamiliar with the methods/related works in this paper.**

This paper proposed a model-based method for offline RL, based on moment matching. Improved numerical results are presented.

**Strengths:**

The method of moment matching is quite novel, which aims to improve the accuracy of the first two moment estimation.

**Weaknesses:**

I am quite confused about Sec 4. Why the Gaussian RVs are related?

I feel the presentation of the paper can be improved.

**Questions:**

The authors mention that the uncertainty-based approach can be overly pessimistic, but some recent paper [1,2] are showing that the uncertainty-based methods achieve the minimax optimal sample complexity, under the tabular setting. Can you explain the reason of such a statement?
[1]Li, Gen, et al. "Settling the sample complexity of model-based offline reinforcement learning." The Annals of Statistics 52.1 (2024): 233-260.
[2]Wang, Yue, Jinjun Xiong, and Shaofeng Zou. "Achieving the Asymptotically Optimal Sample Complexity of Offline Reinforcement Learning: A DRO-Based Approach."

---

> ### Author Rebuttal · Authors · 2024-08-07
>
> Thank you for your thorough reading despite the unfamiliarity.
>
> > I feel the presentation of the paper can be improved.
>
> Given the overall feedback provided via the reviews, we updated the section title and presentation for Section 3, and the _discussion and results_ paragraph in Section 5 (see, e.g., answer to BmgH). Please let us know if further parts of the paper could benefit from presentation improvements in your opinion.
>
> > Why the Gaussian RVs are related?
>
> Assuming that the overall input follows a normal distribution is a common one in the literature coming from the environmental model.
> Assuming in turn that the pre-activation output of a neural layer is also Gaussian can be justified via a central limit theorem argumentation. This is because the input to a specific neuron is a sum of random variables, i.e., the central limit theorem holds and converges to a Gaussian as the width increases. Additionally, if the computation of the first two moments is analytically tractable, the Gaussian distribution has the property of being the maximum entropy distribution, i.e., it imposes the least additional number of assumptions.
> The fact that Algorithm 1 ignores covariance terms is common practice in moment-matching approaches (see the references in the paper), as the computational complexity and memory cost scales quadratically in the number of neurons otherwise.
>
> > The authors mention that the uncertainty-based approach can be overly pessimistic, but some recent papers [1,2] are showing that the uncertainty-based methods achieve the minimax optimal sample complexity, under the tabular setting.
>
> Pessimism is required, however, current uncertainty-based approaches, e.g., MOPO and MOBILE, penalize too much. We do not contradict the two references, but show that our uncertainty estimator is tighter than our baselines'. See the additional experiments we provide in the main rebuttal answer.
> Note that the the term "samples" in those references refers to the size of the underlying data set, see, e.g., Section 2.2 in Li et al. (2024).
>
> _____
> Li et al., _Settling the sample complexity of model-based offline reinforcement learning_ (Annals of Statistics, 2024)

---

> > ### Comment · Reviewer_czcW · 2024-08-13
> >
> > Thank you for the response. The experiments do show some improvement, but I am still not convienced that the previous methods are too pessimistic.
> >
> > Considering the different counstruction of [Li et al. 2024] and the MOPO baseline, I am wondering if the MOPO is the SOTA of current uncertainty-based approaches.

---

> > > ### Author Response · Authors · 2024-08-13
> > > **Clarifications about Pessimism and SOTA**
> > >
> > > Thank you indeed for your response.
> > >
> > > > Thank you for the response. The experiments do show some improvement, but I am still not convienced that the previous methods are too pessimistic.
> > >
> > > We believe the tightness scores we reported in the global response provide direct and strong evidence in favor of our claim that MOMBO applies less penalty than both MOPO and MOBILE. The experiments demonstrate that MOPO applies significantly higher penalty than MOMBO and MOBILE, which is an expected consequence of Theorem 3.6 of Sun et al. (2023). To remind, tightness score measures the difference between the penalty applied by a model and the actual error the Bellman estimator makes. See our global response for details. If there still remains an open issue regarding the relative pessimism of the models in comparison, we are happy to discuss further.
> > >
> > > > Considering the different counstruction of [Li et al. 2024] and the MOPO baseline, I am wondering if the MOPO is the SOTA of current uncertainty-based approaches.
> > >
> > > We actually consider MOBILE as the SOTA of the model-based offline RL models, not MOPO, given its publication date (ICML, 2023) and the lack of a widely known method that improves its performance using commensurate resources. That said, MOPO is still a reference model representing the option of reward penalization based on environment dynamics. MOBILE improves on MOPO by deriving a penalty term based on a sampling-based estimate of the Bellman target. Our contribution is to identify and characterize the problems this sampling-based approach causes and present a novel solution that addresses these problems both theoretically (see our Theorems 1 and 2, and comparison paragraph between lines 247-258.) and empirically (see our AUC results in Table 1 that demonstrate faster convergence).
> > >
> > >
> > >
> > > -----
> > > Sun et al., _Model-bellman inconsistency for model-based offline reinforcement learning_ (ICML 2023)

---

> > > > ### Comment · Reviewer_czcW · 2024-08-13
> > > >
> > > > Thanks for your detailed response! I’m hence keeping my positive rating.

---

### Official Review · Reviewer_3Ppu · 2024-07-13

**Soundness:** 3
**Presentation:** 3
**Contribution:** 3
**Rating:** 7
**Confidence:** 4

**Summary:**

This work addresses the issue of sampling for uncertainty propagation that is the standard practice in offline RL and identifies the high variance of sampling-based estimates as an obstacle to better performance of uncertainty-aware offline RL methods. As an alternative, the authors propose propagating uncertainty with moment matching, which is a deterministic procedure. The method involves propagating the uncertainty of state transitions into the value function where the value function is parameterized with a ReLU network. The method, coined MOMBO, is further theoretically supported with tight bounds on value error approximation, and experiments on the standard Mujoco benchmarks indicate that MOMBO is on par or exceed existing approaches in offline RL.

**Strengths:**

- Uncertainty propagation in RL is a relevant problem to which there is not yet a definitive solution, mainly due to the interplay between uncertainty in state transition and uncertainty in value estimation. This work addresses this issue and proposes a method which is well motivated and theoretically supported. While the method itself does not touch upon the method of uncertainty quantification, rather only the propagation of uncertainty, I believe further investigation into the effect of various UQ methods when used with this propagation could be interesting.
- Empirical results are fairly strong
- Overall the paper is well written

**Weaknesses:**

While the authors conjecture that sampling based approaches to uncertainty propagation has inherent flaws, it would be nice to see more evidence, e.g. empirical evidence that digs into this phenomenon.
As such, the proposed method does not seem like a fundamentally novel approach, rather an improvement in estimation of a given framework. Still, the suggestion of moment matching as a scalable and efficient method of propagating uncertainty seems useful to the community and this paper would fit well alongside existing work.

**Questions:**

N/A

---

> ### Author Rebuttal · Authors · 2024-08-07
>
> Thank you for your positive feedback.
>
> > While the authors conjecture that sampling based approaches to uncertainty propagation has inherent flaws, it would be nice to see more evidence, e.g. empirical evidence that digs into this phenomenon
>
> We include an additional evaluation in our general answer above, to demonstrate both our improvements in accuracy as well as tightness.

---

> > ### Comment · Reviewer_3Ppu · 2024-08-14
> >
> > Thank you for the additional experiments. I maintain my original rating, thank you.

---

### Author Rebuttal · Authors · 2024-08-07

We thank all reviewers for their feedback.

We address all the reviewer comments in our individual responses. To summarize the main changes during the rebuttal phase:
- We conducted an experiment to compare the quality of uncertainty quantifiers among the baselines and our MOMBO, which supports our thesis. _(see below)_
- We provide additional results on a new benchmark (Adroit). _(see below)_
- We reorganized Section 3, _discussion and results_ paragraph of Section 5, and contributions paragraph. _(see, e.g., answer to BmgH)_
## Experiment on Uncertainty Quantifiers
Per the reviewers' request, we conducted an experiment to provide empirical evidence that MOMBO offers a tighter estimation of the ideal uncertainty. To achieve this, we followed the definition of the $\xi$-uncertainty quantifier (see Definition 4.1 in Jin et al. (2021)):
$$| \hat{\mathcal{T}}\hat{Q}(s, a) - \mathcal{T}\hat{Q}(s, a) | \leq \beta \cdot u(s, a)$$
for all $(s, a) \in \mathcal{S} \times \mathcal{A}$.
We aimed to measure the quality of uncertainty penalizers and determine their tightness in model-based offline reinforcement learning algorithms, specifically comparing MOPO, MOBILE, and our algorithm, MOMBO. To achieve this, we report two scores:
1. **Accuracy**: For a given state-action pair, we check whether the equation above holds or not. This measures the quality of the uncertainty quantifier.
2. **Tightness score**: For a given state-action pair we calculate
$$\text{Tightness score}= \beta \cdot u(s, a) - | \hat{\mathcal{T}}\hat{Q}(s, a) - \mathcal{T}\hat{Q}(s, a) |.$$
This score measures how tightly the uncertainty quantifiers can estimate the possible errors.

Here, $\beta \cdot u(s, a)$ represents the penalty applied by the algorithm for a specific state-action pair. $\hat{\mathcal{T}}\hat{Q}(s, a)$ is the learned estimated Bellman target, calculated using learned dynamics, while $\mathcal{T}\hat{Q}(s, a) = r(s, a) + \gamma E_{s' \sim T} [E_{a' \sim \pi}[Q(s', a')]]$ is the true Bellman target. The true Bellman target requires calculating the real expectation for the next state and next action. Since the environments we used are deterministic, we used the actual next state from the environment dynamics and the actual reward. For the expectation of the next action, we used a large number of samples to approximate it accurately.
### Experiment Procedure:
We load the learned policies and dynamics and generate 10 episode rollouts using the real environment in evaluation mode. From these rollouts, we select state, action, reward, and next state tuples at every 10th step, including the final step. For each of these tuples, we calculate the mean accuracy and tightness scores. This process is repeated for each of the 4 seeds across every task, and we report the mean and standard deviation of the accuracies and tightness scores.
### Experiment Results
#### Accuracy
| | MOPO | MOBILE | MOMBO |
|---:|:---:|:---:|:---:|
| halfcheetah-random | 0.02 ± 0.01 | **0.25 ± 0.02** | 0.24 ± 0.07 |
| hopper-random | 0.04 ± 0.01 | **0.85 ± 0.03** | 0.62 ± 0.07 |
| walker2d-random | 0.0 ± 0.0 | 0.55 ± 0.04 | **0.74 ± 0.06** |
| halfcheetah-medium | 0.05 ± 0.01 | 0.25 ± 0.0 | **0.34 ± 0.04** |
| hopper-medium | 0.04 ± 0.01 | 0.41 ± 0.01 | **0.55 ± 0.03** |
| walker2d-medium | 0.02 ± 0.0 | 0.16 ± 0.01 | **0.38 ± 0.02** |
| halfcheetah-medium-replay | 0.08 ± 0.01 | 0.04 ± 0.0 | **0.16 ± 0.0** |
| hopper-medium-replay | 0.02 ± 0.01 | 0.03 ± 0.01 | **0.17 ± 0.01** |
| walker2d-medium-replay | 0.08 ± 0.01 | 0.14 ± 0.01 | **0.36 ± 0.02** |
| halfcheetah-medium-expert | 0.11 ± 0.01 | 0.36 ± 0.03 | **0.44 ± 0.03** |
| hopper-medium-expert | 0.04 ± 0.02 | 0.43 ± 0.02 | **0.51 ± 0.04** |
| walker2d-medium-expert | 0.05 ± 0.01 | **0.47 ± 0.02** | 0.45 ± 0.02 |
#### Tightness Score
| | MOPO | MOBILE | MOMBO |
|---:|:---:|:---:|:---:|
| halfcheetah-random | -14.92 ± 1.14 | -6.88 ± 0.61 | **-6.21 ± 0.28** |
| hopper-random | -1.81 ± 0.16 | -0.64 ± 0.2 | **-0.31 ± 0.65** |
| walker2d-random | -98521227.61 ± 91264317.73 | -0.27 ± 0.08 | **-0.14 ± 0.02** |
| halfcheetah-medium | -15.76 ± 0.66 | -10.03 ± 0.53 | **-9.06 ± 0.26** |
| hopper-medium | -4.08 ± 1.16 | -3.14 ± 0.08 | **-1.3 ± 0.21** |
| walker2d-medium | -8.98 ± 0.63 | -5.02 ± 0.52 | **-4.03 ± 0.24** |
| halfcheetah-medium-replay | -12.16 ± 1.18 | -11.85 ± 0.41 | **-10.4 ± 0.66** |
| hopper-medium-replay | -5.45 ± 0.59 | -3.4 ± 0.04 | **-3.17 ± 0.04** |
| walker2d-medium-replay | -4.45 ± 0.43 | -4.56 ± 0.13 | **-3.78 ± 0.39** |
| halfcheetah-medium-expert | -21.45 ± 2.79 | -13.22 ± 0.47 | **-11.88 ± 0.5** |
| hopper-medium-expert | -7.12 ± 2.6 | -3.5 ± 0.03 | **-3.38 ± 0.02** |
| walker2d-medium-expert | -9.72 ± 0.24 | -5.52 ± 0.36 | **-5.29 ± 0.14** |
### Discussion on Results
To summarize the experiment results: MOMBO provides improved uncertainty estimates for the Bellman target on almost all data sets (9/12 in terms of accuracy, 12/12 in terms of tightness). This improved estimation leads to faster convergence due to clearer gradient signals and results in higher final performance.
## Adroit
Per the reviewer igqb's suggestion, we conducted an experiment on the Adroit domain of the D4RL dataset, specifically reporting on 'pen-human-v1' and 'pen-cloned-v1'. We skipped the other tasks because MOBILE does not provide their hyperparameters, and conducting an extensive hyperparameter search requires too much time, which we do not have for the rebuttal period. We report the MOPO and MOBILE results from the MOBILE paper.
Even without detailed hyperparameter tuning, we are competitive with MOBILE. (MOBILE provides hyperparameters in the appendix, however, these do not agree with those reported in their repository. We rely on the latter.)
| | MOPO | MOBILE | MOMBO |
|---:|:---:|:---:|:---:|
| pen-human | 10.7 | 30.1 ± 14.6 | **32.0 ± 10.7** |
| pen-cloned | 54.6 | **69.0 ± 9.3** | 63.5 ± 19.5 |
_____
Jin et al., _Is Pessimism Provably Efficient for Offline RL?_ (ICML,  2021)

---

### Comment · Area_Chair_xd1N · 2024-08-08
**Discussion Period**

Hi all! Just your friendly area chair checking in.

First, thanks to the reviewers for your work so far on this paper. The discussion period has begun, which includes both reviewers and authors. This is our opportunity to clear up misunderstandings, get answers to questions, and generally gather enough information to make an informed decision grounded in the acceptance criteria.

It seems there is general agreement that the MOMBO approach is novel and interesting. The main critiques raised seem to concern whether there is clear evidence of meaningful empirical advantages of MOMBO over existing methods, and some clarity issues in the manuscript. Does that seem like a fair high-level summary?

Reviewers: please carefully read the other reviews and the authors' responses and let us know what follow-up questions you have and to what extent your evaluation might change as a result of the additional context. Please especially raise any points of disagreement with other reviewers or the authors, as these are opportunities for us to clarify misunderstandings and detect misconceptions.

---

### Decision · Program_Chairs · 2024-09-25

**Decision:**

Accept (poster)

**Comment:**

There is some variation in the overall assessments of the reviews, with most reviewers at least leaning toward acceptance and one reviewer firmly on the reject side. Because of this, in my meta-review I will do my best to synthesize key points from the discussion and corroborate with my own read of the paper.

There is general agreement amongst the reviewers that the application of moment-matching to improve uncertainty estimates is interesting and potentially impactful. They also generally appreciated the theoretical findings that support the hypothesis that moment-matching may provide benefits over Monte Carlo estimation.

The quality of the presentation was somewhat in dispute, with some reviewers expressing that the paper was clear and easy to follow and others expressing concerns about clarity. In my own reading, I did not find clarity of presentation to be a significant barrier to understanding.

Some of the reviewers expressed uncertainty about whether the empirical results sufficiently supported the hypotheses that Monte Carlo estimates were overly conservative and that tighter uncertainty estimates would result in practical benefits. Many of these questions were resolved in the discussion period.

For instance, the authors clarified that, while MOMBO and MOBILE perform similarly after many training steps (and that this was to be expected), the AUC comparison indicates that MOMBO generally outperformed MOBILE for smaller numbers of training steps. I tend to agree with the reviewers that this point was not especially clear in the paper and that this issue was exacerbated by the incorrect use of the term "sample efficiency," rather than the real claim of faster improvement/convergence. The improvements proposed by the authors to address this confusion will improve the paper.

There is still a lingering question of how much faster improvement actually matters in the offline setting, where the central premise is that computational resources are far cheaper than data collection. The authors make the point that it is not necessarily easy to tell when to stop training and that therefore faster improvement implies better performance if training is stopped too early. However, this is a somewhat conjectural claim. There is a long history of research on stopping conditions for neural network training, albeit largely in the supervised learning setting. It's not clear that improving convergence rate is the best or only way to address this issue; that seems like a claim that requires more study. All that said, the empirical findings do at least corroborate the stated intuition and theoretical claims and faster convergence is pretty clearly desirable, even if it's not clear exactly how critical it is.

There were also questions about whether the performance results could be clearly connected to the central hypothesis that MC estimates were overly conservative. The additional experimental results offered by the authors seem to compellingly demonstrate that, indeed, the moment-matching estimates are tighter than the MC estimates. These results seem fairly straightforward to incorporate as they could reside in an appendix with a brief summary of the findings in the main text.

Overall, it seems to me that the paper presents a novel, conceptually sound, well-analyzed, and potentially impactful approach to uncertainty estimation in MBRL and is thus acceptable under the NeurIPS guidelines. It does seem that there is room for improvement in the area of significance, with more work to be done to determine the extent of the practical importance of the theoretical and algorithmic insight. For that reason, I think it's fair for this paper to be considered somewhat borderline. If the paper is ultimately not accepted, I recommend that the authors attend to that particular issue as they continue this line of work.